# MGD³: Mode-Guided Dataset Distillation using Diffusion Models

**Jeffrey A. Chan-Santiago** [1]  **Praveen Tirupattur** [1]  **Gaurav Kumar Nayak** [2]  **Gaowen Liu** [3]  **Mubarak Shah** [1]

## Abstract

Dataset distillation has emerged as an effective strategy, significantly reducing training costs and facilitating more efficient model deployment. Recent advances have leveraged generative models to distill datasets by capturing the underlying data distribution. Unfortunately, existing methods require model fine-tuning with distillation losses to encourage diversity and representativeness. However, these methods do not guarantee sample diversity, limiting their performance. We propose a mode-guided diffusion model leveraging a pre-trained diffusion model without the need to fine-tune with distillation losses. Our approach addresses dataset diversity in three stages: Mode Discovery to identify distinct data modes, Mode Guidance to enhance intra-class diversity, and Stop Guidance to mitigate artifacts in synthetic samples that affect performance. Our approach outperforms state-of-the-art methods, achieving accuracy gains of 4.4%, 2.9%, 1.6%, and 1.6% on ImageNette, ImageIDC, ImageNet-100, and ImageNet-1K, respectively. Our method eliminates the need for fine-tuning diffusion models with distillation losses, significantly reducing computational costs. Our code is available on the project webpage: https://jachansantiago.github.io/mode-guided-distillation/

## 1. Introduction

The rapid advancements in machine learning are marked by a trend towards increasingly large datasets and models to achieve state-of-the-art performance. However, this trend

[1]Center for Research in Computer Vision, University of Central Florida, Orlando, Florida, United States [2]Mehta Family School of Data Science and Artificial Intelligence, Indian Institute of Technology Roorkee, Roorkee, Uttarakhand, India [3]Cisco Research, San Jose, California, United States. Correspondence to: Jeffrey A. Chan-Santiago <jeffrey.chansantiago@ucf.edu>.

*Proceedings of the 42ⁿᵈ International Conference on Machine Learning*, Vancouver, Canada. PMLR 267, 2025. Copyright 2025 by the author(s).

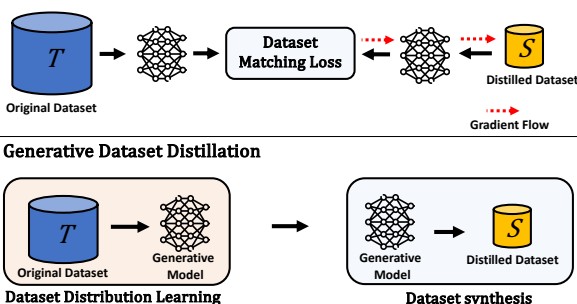

Figure 1. **Optimization-based Dataset Distillation:** Optimizes the distilled dataset to match the statistics of gradient/features of the Original Dataset. **Generative Dataset Distillation:** First, it learns the dataset distribution of the original dataset and then sample a dataset that approximates the original dataset distribution.

presents significant challenges for researchers constrained by limited computation and storage resources. In response, the research community started to focus on developing techniques to address these limitations. While model pruning (Liu et al., 2017; He et al., 2019; Ding et al., 2019; Sharma & Foroosh, 2022) and quantization (Wu et al., 2016; Chen et al., 2021; Chauhan et al., 2023; Xu et al., 2023) have been introduced to improve model efficiency, core set selection and dataset distillation (Wang et al., 2018; Liu et al., 2022) have emerged as prominent techniques for reducing the size of training datasets to accelerate model training.

The process of reducing the training dataset involves removing redundant information while retaining essential data. Core set selection (Welling, 2009; Chen et al., 2010; Rebuffi et al., 2017; Castro et al., 2018) based approaches were initially introduced for building condensed datasets, which involves selecting a few prototypical examples from the original dataset to build the smaller dataset. However, these approaches are limited to choosing the samples from the original dataset, which considerably restricts the expressiveness of the condensed dataset. The task of dataset distillation is to distill information from a large training dataset into a smaller dataset with few synthetic samples such that a model trained on the smaller dataset achieves performance comparable to the model trained on the original dataset.

Optimization-based dataset distillation methods follow the data matching framework (Cazenavette et al., 2022; 2023; Zhao & Bilen, 2023), where the distilled dataset is updated

to mimic the influence of the original dataset when training (see the top of Fig. 1). These methods minimize the distribution gap between the original and distilled datasets by considering different aspects, such as model parameters, long-range training trajectories, or feature distribution. However, these methods are far from optimal, as they need to repeat the execution of their method to synthesize distilled datasets of different sizes. In addition, they tend to generate out-of-distribution samples.

To address these challenges, generative dataset distillation methods (Wang et al., 2023; Zhang et al., 2023; Su et al., 2024) propose storing the knowledge of the dataset into the parameters of a generative model instead of directly condensing it into a smaller synthetic set (see the bottom of Fig. 1). Once trained, the same generative model can generate synthetic datasets of varied sizes. This typically, is achieved by training the generative model with representative and diversity losses.

Among the generative models, diffusion models (Ho et al., 2020) are known for their impressive capabilities in image synthesis. These models achieve perceptual quality comparable to GANs while offering higher distribution coverage, as evidenced by (Dhariwal & Nichol, 2021b). However, they tend to concentrate on denser regions (modes) of the data distribution, resulting in a synthetic dataset that, while representative, often lacks the full diversity of the original data (Gu et al., 2024) (refer to Fig. 2a). Previous works (Gu et al., 2024) address this by explicitly fine-tuning the model with representative and diversity losses to generate representative and diverse samples. With this fine-tuning, the samples are more likely to be generated from different modes of a class (See Fig. 2b). However, this approach requires additional training, which can be computationally expensive and limit its practicality in resource-constrained settings.

We propose a novel approach that extracts diverse and representative samples from a pre-trained diffusion model trained on the target dataset, without additional training or fine-tuning. Our method first estimates prevalent data modes in the **Mode Discovery** stage. Then, diversity is ensured by guiding each sample to a different mode with **Mode Guidance**. However, guiding samples to modes may reduce quality, so we introduce **Stop Guidance** to preserve synthetic data quality (see Fig. 2c). In summary, the key contributions are:

- A novel dataset distillation approach leveraging a pre-trained diffusion model without retraining or fine-tuning.

- Improved diversity and representativeness compared to previous diffusion-based methods.

- Matching or surpassing state-of-the-art results on multiple benchmarks while reducing computational cost.

## 2. Related Work

Dataset distillation has received increased interest in recent years due to its applications in continual learning (Zhao et al., 2021; Zhao & Bilen, 2021; 2023), privacy-preserving datasets (Li et al., 2020; Sucholutsky & Schonlau, 2021), neural architecture search (Zhao et al., 2021; Zhao & Bilen, 2021), and model explainability (Loo et al., 2022). Prior works have explored the problem of dataset distillation and shown how challenging it is to encapsulate datasets in a limited set of examples. Initially, this task was approached using non-generative models, then with generative priors, and more recently with generative models and with decoupled dataset distillation. Below, we discuss works belonging to these categories in detail.

**Non-generative Dataset Distillation Methods.** Dataset distillation condenses information from a large dataset into a smaller one with synthetic images, enabling model training on the smaller dataset with performance comparable to the full dataset. Initially, Zhao et al. (2021) proposed gradient matching to align the model's gradient trained on synthetic data with that on the original dataset. However, this bi-level optimization approach was time-consuming and unscalable. Further advances included feature matching (Zhao & Bilen, 2023), which improved efficiency by removing dependence on bi-level optimization. Later, Cazenavette et al. (2022) proposed long-range matching by matching training trajectories (MTT), optimizing network parameters over multiple training iterations to better synthesize relevant features for updates.

**Dataset Distillation with Generative Priors.** Recent advancements have introduced generative priors into the optimization process. GAN-IT (Zhao & Bilen, 2022) shifted the focus from the pixel space to latent codes of pre-trained GANs, optimizing these codes rather than working directly in image space. GLaD (Cazenavette et al., 2023) built on this by incorporating generative priors with StyleGAN for high-resolution datasets, yielding images that more closely match the dataset distribution and improve performance. H-GLaD (Zhong et al., 2024) further enhanced this by focusing on deeper feature layers for hierarchical optimization. Additionally, LD3M (Moser et al., 2024) utilized a latent diffusion model to optimize synthetic datasets directly in the model's latent space, improving performance by refining latent codes through the denoising and diffusion processes. Despite their success on small-resolution datasets, these methods struggle with high-resolution datasets (e.g., $256 \times 256$, 20 images per class), often being computationally expensive and less efficient, leading to the emergence of more effective distillation methods from generative models.

**Generative Dataset Distillation Methods.** Recent works (Zhang et al., 2023; Gu et al., 2024; Su et al., 2024) have explored dataset distillation via generative models, moving

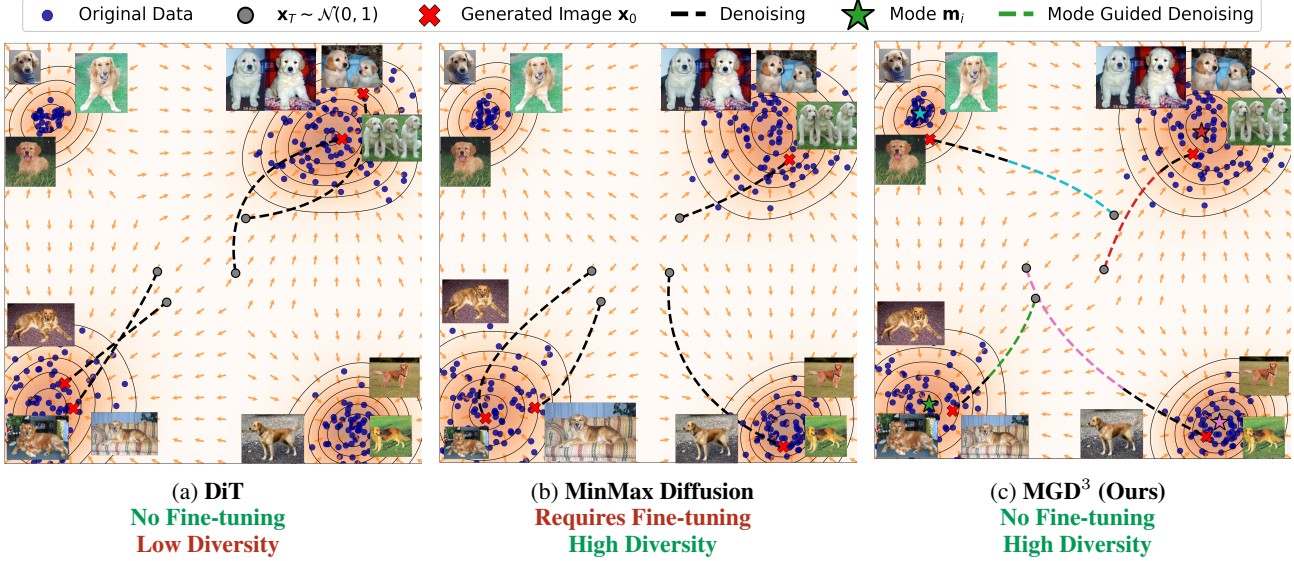

(a) **DiT**
No Fine-tuning
Low Diversity

(b) **MinMax Diffusion**
Requires Fine-tuning
High Diversity

(c) **MGD³ (Ours)**
No Fine-tuning
High Diversity

*Figure 2.* Overview of the gradient field (score function) during the denoising process in latent diffusion for a specific class $c$. The original data distribution, marked by blue dots, shows denser regions (orange shadow) in the gradient field. To generate an image $\hat{X}_i$, noise $x_T{}^i \sim N(0, \mathbf{I})$ is sampled. In (a), a pre-trained diffusion model demonstrates imbalanced mode likelihood, leading to limited sample diversity and repeated modes. (b) shows MinMax Diffusion, which fine-tunes the model to enhance diversity by balancing mode likelihoods, but still faces redundancies based on initial noise conditions. (c), the proposed method introduces mode guidance in the denoising process (green and red traces), directing samples towards distinct modes (stars). After $k$ steps of guidance, it transitions to unguided denoising (black trace), achieving high diversity and consistency without the need for fine-tuning.

beyond using generative priors merely to optimize latent codes. Instead, generative dataset distillation trains models to synthesize entire distilled datasets. Zhang et al. (2023) introduced a class-conditional GAN with a learnable codebook per image, optimized using multiple losses for realism, representativeness, and diversity. Gu et al. (2024) extended this to diffusion models by fine-tuning a pretrained model with representative and diversity losses. Su et al. (2024) proposed $D^4M$, which uses Stable Diffusion and replaces random noise with noisy modes during sampling; however, early denoising noise often leads to limited diversity and representativeness. While prior methods rely on complex loss designs and additional training, we propose a training-free approach that achieves both representativeness and diversity without such overhead.

**Decoupled Dataset Distillation.** Recent advances in dataset distillation have introduced decoupled formulations that scale to ImageNet. Yin et al. (2023) proposed SRe²L, a Squeeze-Recover-Relabel framework that: (1) squeezes dataset statistics into a model through training, (2) recovers information by optimizing synthetic data to match batch-norm statistics, and (3) boosts performance via soft labels from a pretrained model. Extending this, Shao et al. (2024a) introduced G-VBSM, applying statistical matching to convolutional layers with multi-backbone support, achieving state-of-the-art results from CIFAR-100 to ImageNet-1K. To improve sample fidelity, Sun et al. (2024) proposed a fast, diversity-driven method, distilling ImageNet-1K into 10 images per class within minutes. Shao et al. (2024b) further explored the design space, introducing soft category-aware matching and optimization strategies such as small batches and adaptive learning rates. In contrast to methods that use discriminative models and image optimization—often producing artifacts and poorly aligned samples—we train a generative model to encode the data distribution and recover samples via guided sampling, yielding results more consistent with the original dataset.

## 3. Preliminaries

**Dataset Distillation:** Given a large-scale dataset with the training set $\mathcal{T} = \{(X_i, y_i)\}_{i=1}^{N_\mathcal{T}}$, the goal of dataset distillation is to build a smaller synthetic dataset $\mathcal{S} = \{(\tilde{X}_i, \tilde{y}_i)\}_{i=1}^{N_\mathcal{S}}$, where $N_\mathcal{S} << N_\mathcal{T}$ and $X_i, \tilde{X}_i$ are the original and synthetic images with the corresponding class labels $y_i, \tilde{y}_i$. In addition, the model $\phi_\mathcal{T}$ trained on the original training set should achieve similar test performance as the model $\phi_\mathcal{S}$ trained on the smaller synthetic dataset; i.e. if $\mathcal{A}$ is the accuracy of a model on the test set ($\mathcal{T}_e$), then $\mathcal{A}(\phi_\mathcal{T}) \sim \mathcal{A}(\phi_\mathcal{S})$. During the evaluation, the size of the distilled dataset $N_\mathcal{S}$, is set based on the distillation budget, denoted by IPC, the number of images allocated per class.

Our approach builds on the foundations of prior generative models, such as Gu et al. (2024); Su et al. (2024); Zhang

et al. (2023), which address dataset distillation by approximating the dataset distribution through sampling diverse and representative instances. This line of work can be characterized as dataset distillation through dataset matching. Where the objective of the data distillation is defined as

$$\left\| \mathbb{E}_{x \sim P(\mathcal{D})} \big[ \ell(\phi_{\mathcal{T}}(x), y) \big] - \mathbb{E}_{x \sim P(\mathcal{D})} \big[ \ell(\phi_{\mathcal{S}}(x), y) \big] \right\| < \epsilon$$

where $P(\mathcal{D})$ denotes the real data distribution, and $\ell$ is a loss function. Note that this formulation is similar to the coreset methods. However, the use of generative models is more flexible because it's not limited to only choosing original samples.

**Diffusion Model:** The denoising probabilistic diffusion model (DDPM) is a generative model, $\mathcal{G}$, that learns a mapping between Gaussian noise and the data distribution through a series of T denoising steps. $\mathcal{G}$ assumes a Markov chain that gradually adds noise to a sample $x_0$ in the data distribution, which is called the forward process. The forward process of $\mathcal{G}$ is defined as $q(x_t|x_{t-1}) = N(\sqrt{1 - \beta_t} x_{t-1}, \beta_t \mathbf{I})$, where $\beta_t$ is the variance schedule for the time step t. In practice, this is done using the reparametrization trick $x_t = \sqrt{\hat{\alpha}} x_0 + \sqrt{1 - \hat{\alpha}} \epsilon_t$, where $\epsilon_t \sim N(0, \mathbf{I})$.

Image generation is done by the reverse process of $\mathcal{G}$, where $\epsilon_\theta$ is the noise prediction network, trained to reverse the Markov chain $p_\theta(x_{t-1}|x_t) = N(\mu_\theta(x_t), \Sigma_\theta(x_t))$, where $\theta$ corresponds to the parameters of the model and $\mu_\theta(x_t), \Sigma_\theta(x_t)$ are the $\mu$ and $\Sigma$ predictions of the denoising models. $\mu_\theta(x_t)$ is computed as follows:

$$\mu_\theta(x_t) = \frac{1}{\sqrt{1 - \beta_t}} \left( x_t - \frac{1}{\sqrt{1 - \alpha}} \epsilon_\theta(x_t, t) \right) + \sigma_t \mathbf{z} \quad (1)$$

where $\mathbf{z} \sim N(0, 1)$ and $\sigma_t$ is the variance schedule. $\epsilon_\theta(x_t, t)$ is the output of the noise prediction network that is trained to predict the added noise with the simple loss defined as

$$\mathcal{L}_\theta = ||\epsilon_\theta(x_t, t) - \epsilon_t||^2. \quad (2)$$

After training, $\mathcal{G}$ can generate samples by sampling from the noise distribution and running the reverse process. In this work, we use a class-conditioned diffusion model $\mathcal{G}_c$, where the output of the noise prediction network conditioned with the class $c$, is denoted as $\epsilon_\theta(x_t, t, c)$.

**Diffusion Guidance:** The sampling process of DDPM is equivalent to score-based generative models by interpreting $\epsilon_\theta(x_t, t) = -\sqrt{\alpha} \nabla_x \log p(x_t)$, where $\nabla_x \log p(x_t)$ is an estimation of the score function. For the case of class-conditioned generation, by using Bayes' rule the score function can be derived as:

$$\nabla_x \log p(x_t|c) = \nabla_x \log p(x_t) + \nabla_x \log p(c|x_t), \quad (3)$$

where $\nabla_x \log p(c|x_t)$ is the gradient of the class-conditional log-likelihood. It's important to note that $\nabla_x \log p(c|x_t)$

represents the drift of the diffusion process towards the distribution of the class $c$. Dhariwal & Nichol (2021a) employ a classifier to estimate the class-conditional log-likelihood and used it as a guidance signal to direct the diffusion process towards the desired class. Later, Ho & Salimans (2021) suggested using a combination of unconditional generation and conditional diffusion (eq. 4) to remove the dependency on the classifier and demonstrated improved results and called this classifier-free guidance. Classifier-free guidance is defined as

$$\tilde{\epsilon}_\theta(x_t, t, c) = (1 - w) \cdot \epsilon_\theta(x_t, t, c) - w \cdot \epsilon_\theta(x_t, t), \quad (4)$$

where the $w$ is the guidance scale that controls how strong the guidance is applied.

## 4. Method

We propose a method for generating diverse and representative class samples by harnessing a diffusion model trained on the target dataset. The core idea is to sample from the denser regions of the data distribution, known as modes, during the reverse process. These modes correspond to clusters of images with similar features and are representative of the class. However, diffusion models often oversample the most prominent modes, which creates redundancies in the distilled dataset, especially when the number of dominant modes for a class is smaller than the desired number of images per class (IPC).

Our three-stage approach, shown in Fig. 3, eliminates the need for fine-tuning while preserving mode diversity. In the first stage, *mode discovery*, we estimate a diverse set of modes for each class in the dataset. The second stage leverages our proposed *mode guidance* to control the reverse process and enable sampling from the estimated mode distribution. During sampling, the guidance is applied until the *stop guidance*—the third stage—is triggered, ensuring control over the quality of the generated samples.

### 4.1. Mode Discovery

In the mode discovery stage, the main objective is to identify the $N$ modes of a specific class in the original dataset distribution. This discovery is performed using the original dataset in the latent space of the VAE encoder ($V_{enc}$). The motivation for this approach is that the generative space captures the overall content of the image rather than discriminative features, which can be limited to specific textures in the image. Any clustering algorithm can be used to estimate the modes for a particular class. In our experiments, we use K-Means centroids, as they are shown to be effective in our ablations with various mode discovery algorithms (see Appendix Section D). Once the modes are identified, our goal is to sample images from these estimated modes.

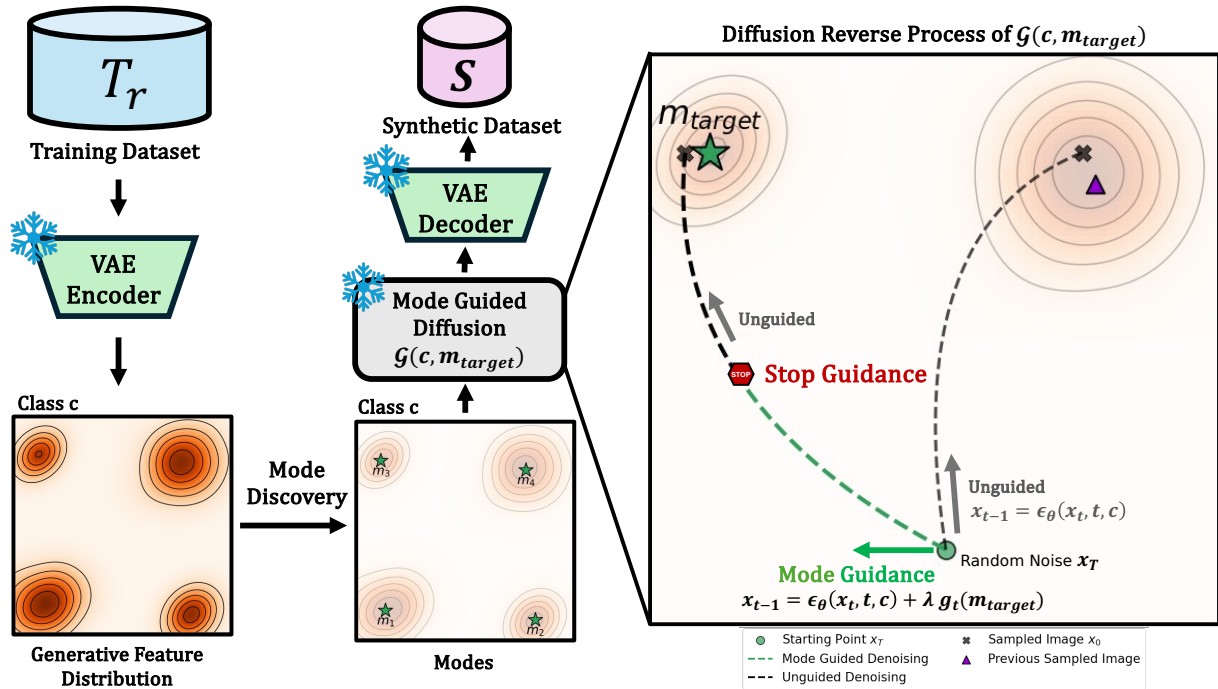

*Figure 3.* Overview of the proposed method for distilled dataset synthesis using a diffusion model. Our approach consists of three key stages: *Mode Discovery*, *Mode Guidance*, and *Stop Guidance*. (**Left**) In the Mode Discovery stage, we estimate the $N$ modes of the original dataset within the generative space of the latent diffusion model. (**Right**) Given a mode $m_{target}$ and a class $c$, the Mode-Guided Diffusion process directs the generation toward the specified mode $m_{target}$. This guidance is applied for $t_{stop}$ steps until the Stop Guidance stage, after which unguided diffusion takes over. During sampling, mode guidance ensures that images from the desired mode $m_k$ are generated using the pre-trained diffusion model. If no guidance is applied, the generation follows the unguided (grey) path, which can lead to redundancies in the dataset.

## 4.2. Mode Guidance

At the image synthesis stage, our goal is to generate high-quality images belonging to a specific class mode. Given a class $c$ and a set of discovered modes for that class denoted as $\mathbf{M_c} = \{m_1, ..., m_N\}$, the mode guidance score is computed for a particular mode $m_i$ using the following equation:

$$\mathbf{g}_t = (m_i - \hat{x_0}^t), \qquad (5)$$

where $\hat{x_0}^t$ is the predicted denoised latent vector at timestep $t$ during the reverse process. We apply this guidance signal at the $x_t$ timestep as follows:

$$\hat{\epsilon}_\theta(x_t, t, c) = \tilde{\epsilon}_\theta(x_t, t, c) + \lambda \cdot \mathbf{g}_t \cdot \sigma_t, \qquad (6)$$

where $\lambda$ is a scalar that controls the strength of the guidance signal.

To synthesize an image from a particular mode $m_i$, in the diffusion model $\mathcal{G}$ the mode guidance score is computed at each iteration of the reverse process using Eq.6. This score represents the direction from the predicted value to the mode $m_i$. The guidance signal is then added to the noise function at the appropriate time step in the diffusion process. By adjusting the strength of the guidance signal, we can regulate the impact of the mode on the generated image.

## 4.3. Stop Guidance

The reverse diffusion process can be divided into three distinct stages: the chaotic stage (first 20%), the semantic stage (20% to 50%), and the refinement stage (final 50%) (Yu et al., 2023). During the refinement stage, mode guidance becomes unnecessary since its primary purpose is to guide the synthetic image towards the mode in the high semantic space. Our initial experiments revealed that maintaining strong guidance towards a particular mode $m_i$ throughout the full reverse process often compromises class fidelity and introduces image artifacts (See Fig. 7b $t_{stop} = 0$). To address these issues, we introduce the stop guidance mechanism, which involves setting the guidance parameter $\lambda$ to zero in Equation 6 when the timestep $t$ falls below a timestep $t_{stop}$ during the reverse process. In the Appendices A and J we examine the effects of different stop guidance timesteps ($t_{stop}$) on image generation quality.

## 5. Experiments

**Datasets and evaluation.** To assess our approach's effectiveness, we thoroughly examine the available benchmarks for distilling high-resolution datasets ($256 \times 256$). The

*Table 1.* Comparison of performance between pre-trained diffusion models and state-of-the-art methods on ImageNet subsets, evaluated using the hard-label protocol. Results are based on ResNet-10 with average pooling, with the best performance highlighted in **bold**. Accuracy is used as the evaluation metric.

| | Nette | | | IDC | | |
|---|---|---|---|---|---|---|
| IPC | 10 | 20 | 50 | 10 | 20 | 50 |
| Random | $54.2_{\pm1.6}$ | $63.5_{\pm0.5}$ | $76.1_{\pm1.1}$ | $48.1_{\pm0.8}$ | $52.5_{\pm0.9}$ | $68.1_{\pm0.7}$ |
| DM (Zhao & Bilen, 2023) | $60.8_{\pm0.6}$ | $66.5_{\pm1.1}$ | $76.2_{\pm0.4}$ | $52.8_{\pm0.5}$ | $58.5_{\pm0.4}$ | $69.1_{\pm0.8}$ |
| MinMaxDiff (Gu et al., 2024) | $62.0_{\pm0.2}$ | $66.8_{\pm0.4}$ | $76.6_{\pm0.2}$ | $53.1_{\pm0.2}$ | $59.0_{\pm0.4}$ | $69.6_{\pm0.2}$ |
| LDM (Rombach et al., 2022) | $60.3_{\pm3.6}$ | $62.0_{\pm2.6}$ | $71.0_{\pm1.4}$ | $50.8_{\pm1.2}$ | $55.1_{\pm2.0}$ | $63.8_{\pm0.4}$ |
| LDM+ Disentangled Diffusion ($D^4M$ (Su et al., 2024)) | $59.1_{\pm0.7}$ | $64.3_{\pm0.5}$ | $70.2_{\pm1.0}$ | $52.3_{\pm2.3}$ | $55.5_{\pm1.2}$ | $62.7_{\pm0.8}$ |
| LDM+ **MGD**$^3$ **(Ours)** | $61.9_{\pm4.1}$ | $65.3_{\pm1.3}$ | $74.2_{\pm0.9}$ | $53.2_{\pm0.2}$ | $58.3_{\pm1.7}$ | $67.2_{\pm1.3}$ |
| DiT (Peebles & Xie, 2023) | $59.1_{\pm0.7}$ | $64.8_{\pm1.2}$ | $73.3_{\pm0.9}$ | $54.1_{\pm0.4}$ | $58.9_{\pm0.2}$ | $64.3_{\pm0.6}$ |
| DiT+ Disentangled Diffusion ($D^4M$ (Su et al., 2024)) | $60.4_{\pm3.4}$ | $65.5_{\pm1.2}$ | $73.8_{\pm1.7}$ | $51.1_{\pm2.4}$ | $58.0_{\pm1.4}$ | $64.1_{\pm2.5}$ |
| DiT + **MGD**$^3$ **(Ours)** | $\mathbf{66.4_{\pm2.4}}$ | $\mathbf{71.2_{\pm0.5}}$ | $\mathbf{79.5_{\pm1.3}}$ | $\mathbf{55.9_{\pm2.1}}$ | $\mathbf{61.9_{\pm0.9}}$ | $\mathbf{72.1_{\pm0.8}}$ |

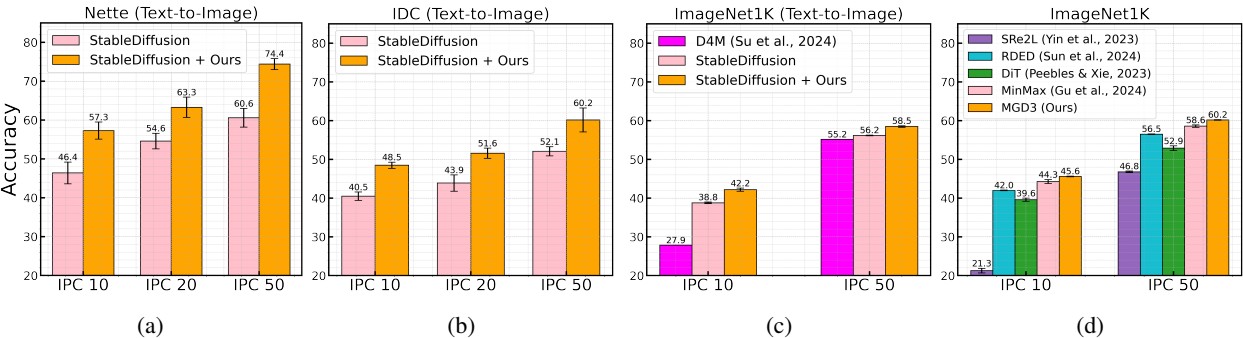

*Figure 4.* Evaluation results across multiple datasets. (a–c) Accuracy of the Text-to-Image model using the soft-label protocol: (a) Nette dataset, (b) IDC dataset, and (c) ImageNet-1K dataset. (d) ImageNet-1K classification accuracy of the DiT + MGD$^3$ model compared to other state-of-the-art (SOTA) methods. All reported values are the mean accuracy over three runs.

datasets we evaluate include ImageNet-1K, ImageNet-100, ImageNetIDC, ImageNette, and ImageNet-A to ImageNet-E. Additionally, we include results from ImageWoof in the Appendix E. We use two protocols for evaluation: a hard-label protocol and a soft-label protocol.

The *hard-label protocol* generates a dataset with its corresponding class labels, trains a network from scratch, and evaluates the network on the original test set. This process is repeated three times for target architectures, and the accuracy mean and standard deviation are reported. Random resize-crop and CutMix are applied as augmentation techniques during the target network's training. For more detailed technical information about the protocol, please refer to Gu et al. (2024). Similar to the existing literature, we evaluate our model in various IPCs ranging from 10 to 100. This protocol was used to evaluate ImageNet-100, ImageNette, and ImageNetIDC datasets.

In *soft-label protocol*, region-based soft-labels are generated with a pre-trained network as proposed by Sun et al. (2024). The region-based soft-labels $y_{i,m}$ are generated as follows: $y_{i,m} = \phi_\mathcal{T}(x_{i,m})$, where $\phi_\mathcal{T}$ is the pretrained model and $x_{i,m}$ is the $m$-th crop of the $i$-th image. When training a

model $\phi_\mathcal{S}$ on the distilled dataset the objective loss is $\mathcal{L} = -\sum_j \sum_m y_{j,m} \log \phi_\mathcal{S}(x_{j,m})$. For ImageNet-1k evaluation, we follow this protocol. Similarly to Sun et al. (2024); Gu et al. (2024), we use ResNet-18 as a teacher and student network architecture for this setup.

**Baselines.** We compare several baselines to contextualize the performance of our method. First, we include the pre-trained DiT XL/2, which represents diffusion models without mode guidance. Second, we evaluate MinMax diffusion with DiT XL/2, where the model is fine-tuned to encourage diversity and representativeness. Additionally, for the ImageNette and IDC datasets, we incorporate a class-conditioned Latent Diffusion Model (LDM) (Rombach et al., 2022) trained on ImageNet-1k. This allows us to compare the U-Net architecture (used in LDM) with the Transformer-based DiT architecture within the diffusion framework. In our experiments, both DiT and LDM by default use the DDPM sampler. Lastly, to enable a fair comparison with $D^4M$ (Su et al., 2024) under our hard label protocol, we apply its disentangled diffusion stage without incorporating the soft labels used in their Training Time Matching procedure on ImageNette and IDC datasets.

*Table 2.* Performance comparison on ImageNet-100. The best results are marked as **bold**.

| | 10 (0.8%) | | | 20 (1.6%) | | |
|---|---|---|---|---|---|---|
| | ConvNet-6 | ResNetAP-10 | ResNet-18 | ConvNet-6 | ResNetAP-10 | ResNet-18 |
| Random | $17.0_{\pm 0.3}$ | $19.1_{\pm 0.4}$ | $17.5_{\pm 0.5}$ | $24.8_{\pm 0.2}$ | $26.7_{\pm 0.5}$ | $25.5_{\pm 0.3}$ |
| Herding (Welling, 2009) | $17.2_{\pm 0.3}$ | $19.8_{\pm 0.3}$ | $16.1_{\pm 0.2}$ | $24.3_{\pm 0.4}$ | $27.6_{\pm 0.1}$ | $24.7_{\pm 0.1}$ |
| IDC-1 (Kim et al., 2022) | $\mathbf{24.3}_{\pm 0.5}$ | $25.7_{\pm 0.1}$ | $\mathbf{25.1}_{\pm 0.2}$ | $28.8_{\pm 0.3}$ | $29.9_{\pm 0.2}$ | $30.2_{\pm 0.2}$ |
| MinMaxDiff (Gu et al., 2024) | $22.3_{\pm 0.5}$ | $24.8_{\pm 0.2}$ | $22.5_{\pm 0.3}$ | $29.3_{\pm 0.4}$ | $32.3_{\pm 0.1}$ | $31.2_{\pm 0.1}$ |
| **MGD$^3$ (Ours)** | $23.4_{\pm 0.9}$ | $\mathbf{25.8}_{\pm 0.5}$ | $23.6_{\pm 0.4}$ | $\mathbf{30.6}_{\pm 0.4}$ | $\mathbf{33.9}_{\pm 1.1}$ | $\mathbf{32.6}_{\pm 0.4}$ |
| Full | $79.9_{\pm 0.4}$ | $80.3_{\pm 0.2}$ | $81.8_{\pm 0.7}$ | $79.9_{\pm 0.4}$ | $80.3_{\pm 0.2}$ | $81.8_{\pm 0.7}$ |

**Text-to-Image Diffusion Model.** Our method is adaptable to various diffusion models, with optimal performance observed when the model is pre-trained on the target dataset. To assess the generalizability of our approach, we test it on a general-purpose diffusion model, specifically a text-to-image diffusion model. This evaluation poses challenges due to the potential mismatch between the model's training data and the target dataset. For this setup, the baseline is Text-to-Image Stable Diffusion model without mode guidance, allowing us to demonstrate the impact of integrating mode guidance in the generated dataset. For sampling, we use the class names as a text prompt.

**Implementation details.** Our pre-trained model $\mathcal{G}$ is DiT-XL/2 trained on ImageNet, and the image size is 256 x 256. We use the sampling strategy described in Peebles & Xie (2023), which uses 50 sampling steps using classifier-free guidance with a guidance scale of $4.0$. For Mode Guidance, we set $\lambda$ to $0.1$, and in our experiments, we use stop guidance $t_{stop} = 25$. We use $K$-means to perform mode discovery; we set $k = IPC$. We use a single NVIDIA RTX A5000 GPU with 24GB VRAM to run our experiments.

### 5.1. Comparison with state-of-the-art methods

We compare our method with current SOTA methods on various image datasets and architectures. Our method significantly outperforms previous approaches across various benchmark datasets and target architectures.

**ImageNette and ImageIDC.** On the ImageNette dataset, our method using DiT achieves notable performance gains of $4.4\%$, $4.4\%$, and $2.9\%$ for IPC values of 10, 20, and 50, respectively, surpassing previous state-of-the-art (SOTA) methods (see Tab. 1). Similarly, on the ImageIDC dataset, our method demonstrates improvements of $2.8\%$, $2.9\%$, and $2.5\%$ for IPC 10, 20, and 50, respectively, outperforming prior SOTA results. Tab. 1 highlights that our approach consistently enhances the performance of DiT and LDM. Furthermore, in the Text-to-Image evaluation mode, our method with guidance surpasses Stable Diffusion on both datasets, as illustrated in Fig. 4a and 4b.

**ImageNet-100 and ImageNet-1K.** Tab. 2 shows comparison to SOTA in ImageNet-100 in IPC 10 and 20 in various target architectures. Our method surpasses the previous SOTA by $1.3\%$, $1.6\%$, and $1.4\%$ in IPC 20 for various target architectures. It also outperformes the MinMax diffusion approach in IPC 10 and achieves the best performance with the ResNetAP-10 target architecture while delivering the second-best results for ConvNet-6 and ResNet-18 architectures. It is important to note that our method is substantially more computationally efficient compared to IDC and MinMax (see Computational Cost below). We also compare our method with SOTA in ImageNet-1K on the soft-label protocol on IPC 10 and 50 in Fig. 4d. Our method achieves SOTA outperforming previous SOTA by $1.3\%$ and $1.6\%$. While using a Text-to-Image diffusion in ImageNet-1k, our method shows an improvement of $3.4\%$ and $2.3\%$ in IPC 10 and IPC 50 over Stable Diffusion as shown in Fig. 4c.

**Performance on Larger Models.** To evaluate the scalability of our approach, we assess its performance on larger backbone architectures—ResNet-50 and ResNet-101—under the IPC50 setting on ImageNet-1k. Table 3 compares our method against several existing approaches across ResNet-18, ResNet-50, and ResNet-101. Our method consistently outperforms prior work on both larger backbones, demonstrating strong generalization to high-capacity models. Notably, while ResNet-18 achieves 69.8% accuracy when trained on the full dataset, our method achieves 86% accuracy using only 3.9% of the data, highlighting both its data efficiency and strong relative performance.

**Computational Cost.** Our method achieves state-of-the-art performance on all datasets, except ImageNet-100, where the best-performing method, IDC-1 (Kim et al., 2022), has slightly better results than ours but with much higher computational cost. For example, MinMax (Gu et al., 2024) took 10 hours to produce a distilled dataset for ImageNet-100 with IPC-10, while IDC-1 (Kim et al., 2022) took over 100 hours for the same. The optimization strategy proposed in IDC-1 (Kim et al., 2022) can not scale up to the ImageNet-1K, and MinMax diffusion requires expensive fine-tuning of the diffusion model, especially for larger datasets like

*Table 3.* Comparison of top-1 accuracy across different methods and backbone architectures (ResNet-18, ResNet-50, ResNet-101) under the IPC50 setting on ImageNet. A dash (–) indicates that the result was not reported.

| Method | ResNet-18 | ResNet-50 | ResNet-101 |
|---|---|---|---|
| *Full Dataset* | *69.8* | *80.9* | *81.9* |
| SR$^2$L (Yin et al., 2023) | $46.8 \pm 0.2$ | $55.6 \pm 0.3$ | $60.8 \pm 0.5$ |
| G-VBSM (Shao et al., 2024a) | $51.8 \pm 0.4$ | $58.7 \pm 0.3$ | $61.0 \pm 0.4$ |
| RDED (Sun et al., 2024) | $56.5 \pm 0.1$ | – | $61.2 \pm 0.4$ |
| EDC (Shao et al., 2024b) | $58.0 \pm 0.2$ | $64.3 \pm 0.2$ | $64.9 \pm 0.2$ |
| D$^4$M (Su et al., 2024) | $55.2 \pm 0.1$ | $62.4 \pm 0.1$ | $63.4 \pm 0.1$ |
| **Ours** | $\mathbf{60.2 \pm 0.1}$ | $\mathbf{64.6 \pm 0.4}$ | $\mathbf{67.7 \pm 0.4}$ |

*Table 4.* Comparison of our method with generative prior methods on ImageNet subsets A to E with IPC-10.

| Distil Alg. | Method | ImNet-A | ImNet-B | ImNet-C | ImNet-D | ImNet-E |
|---|---|---|---|---|---|---|
| DC | Pixel | $52.3_{\pm 0.7}$ | $45.1_{\pm 8.3}$ | $40.1_{\pm 7.6}$ | $36.1_{\pm 0.4}$ | $38.1_{\pm 0.4}$ |
| | GLaD | $53.1_{\pm 1.4}$ | $50.1_{\pm 0.6}$ | $48.9_{\pm 1.1}$ | $38.9_{\pm 1.0}$ | $38.4_{\pm 0.7}$ |
| | H-GLaD | $54.1_{\pm 1.2}$ | $52.0_{\pm 1.1}$ | $49.5_{\pm 0.8}$ | $39.8_{\pm 0.7}$ | $40.1_{\pm 0.7}$ |
| | LM3D | $55.2_{\pm 1.0}$ | $51.8_{\pm 1.4}$ | $49.9_{\pm 1.3}$ | $39.5_{\pm 1.0}$ | $39.0_{\pm 1.3}$ |
| DM | Pixel | $44.4_{\pm 0.5}$ | $52.6_{\pm 0.4}$ | $50.6_{\pm 0.5}$ | $47.5_{\pm 0.7}$ | $35.4_{\pm 0.4}$ |
| | GLaD | $52.8_{\pm 1.0}$ | $51.3_{\pm 0.6}$ | $49.7_{\pm 0.4}$ | $36.4_{\pm 0.4}$ | $38.6_{\pm 0.7}$ |
| | H-GLaD | $55.1_{\pm 0.5}$ | $54.2_{\pm 0.5}$ | $50.8_{\pm 0.4}$ | $37.6_{\pm 0.6}$ | $39.9_{\pm 0.7}$ |
| | LM3D | $57.0_{\pm 1.3}$ | $52.3_{\pm 1.1}$ | $48.2_{\pm 4.9}$ | $39.5_{\pm 1.5}$ | $39.4_{\pm 1.8}$ |
| - | **MGD$^3$ (Ours)** | $\mathbf{63.4_{\pm 0.8}}$ | $\mathbf{66.3_{\pm 1.1}}$ | $\mathbf{58.6_{\pm 1.2}}$ | $\mathbf{46.8_{\pm 0.8}}$ | $\mathbf{51.1_{\pm 1.0}}$ |

ImageNet-1k. In contrast, we use pre-trained diffusion models to create a distilled dataset with no additional computational cost for fine-tuning and minimal overhead for mode discovery. For comparison, our method takes 0.42 hours to generate a synthetic dataset for ImageNet-100 with IPC-10. This highlights the computational efficiency of our model compared to previous approaches.

**Comparison with Generative Prior Methods.** We compare our method against GLaD, H-GLaD, and LM3D in their cross-architecture setup, using AlexNet, VGG11, ResNet18, and ViT for performance evaluation. The evaluation was done by running the evaluation five times per architecture and reporting the mean performance across all the architectures. We evaluate our model in 5 subsets: A, B, C, D, and E of ImageNet. Our method was trained using the hard-label protocol. Tab. 4 shows that our method outperforms previous approaches in this setup. Additionally, these methods face scalability challenges for large datasets such as ImageNet-1K or higher IPC values (>50) due to their high time and space complexity.

### 5.2. Ablation Experiments

**Effect of each component.** To assess the impact of each proposed component, we incrementally evaluated the following: 1) Mode Discovery, 2) Mode Guidance, and 3) Stop Guidance. Mode Discovery involves performing $K$-means per class on the original dataset and selecting the closest sample to the k-means centroid. We conduct the evaluation on the ImageNette dataset with IPC 10, and report the accuracy of ConvNet-6, ResNet10 with average pooling, and ResNet18. Tab. 5 demonstrates that using diffusion with mode guidance enhances mode discovery and that stop guidance is crucial for achieving improved performance.

**Visualizing t-SNE.** To analyze the distilled dataset's coverage, we visualize a t-SNE plot of the distilled dataset from the DiT, MinMax Diffusion, and our method. Fig. 5 illustrates that the DiT distilled dataset is mostly contained in one region of the original dataset distribution, while MinMax Diffusion extends to a broader area of the data distribution.

*Table 5.* Ablation study on the component of our proposed method. The results are on the ImageNette dataset with IPC 10. Each component contributes to the overall performance.

| Test Model | Mode Disc. | Mode Guid. | Stop Guid. | Acc. |
|---|---|---|---|---|
| ConvNet-6 | | | | $53.2_{\pm 1.4}$ |
| ResNetAP-10 | ✔ | - | - | $57.1_{\pm 1.3}$ |
| ResNet-18 | | | | $53.5_{\pm 0.6}$ |
| ConvNet-6 | | | | $57.5_{\pm 1.3}$ |
| ResNetAP-10 | ✔ | ✔ | - | $63.8_{\pm 1.6}$ |
| ResNet-18 | | | | $62.0_{\pm 2.2}$ |
| ConvNet-6 | | | | $\mathbf{59.6_{\pm 2.2}}$ |
| ResNetAP-10 | ✔ | ✔ | ✔ | $\mathbf{66.4_{\pm 2.4}}$ |
| ResNet-18 | | | | $\mathbf{64.4_{\pm 1.9}}$ |

However, the distilled dataset from our method covers a broader area of the data distribution than both methods.

**Representativeness and Diversity.** While t-SNE provides a qualitative visualization of diversity, it does not present the complete picture. We are also interested in representativeness. With this in mind, our goal is to empirically measure diversity and representativeness in the t-SNE space described above. To measure diversity, we calculate the pairwise distance of all samples within a class for the distilled dataset and report the minimum distance per sample. To measure representativeness, we calculate the mean distance to the 50 closest samples in the original dataset, where a greater distance indicates lower representativeness and a smaller distance indicates higher representativeness.

We compare the diversity and representativeness of each class for DiT, MinMax diffusion, and our method as shown in Fig. 6. For clarity in visualization, we plot $1 -$ representativeness, so that higher values indicate higher representativeness. Our experiment indicates that DiT examples show partial representative and partial diversity. On the other hand, MinMax produces more diverse examples than DiT, although some classes lack diversity. Our method demonstrates that our samples are both diverse and representative. Furthermore, we provide additional results about representativeness and diversity in the Appendix B.

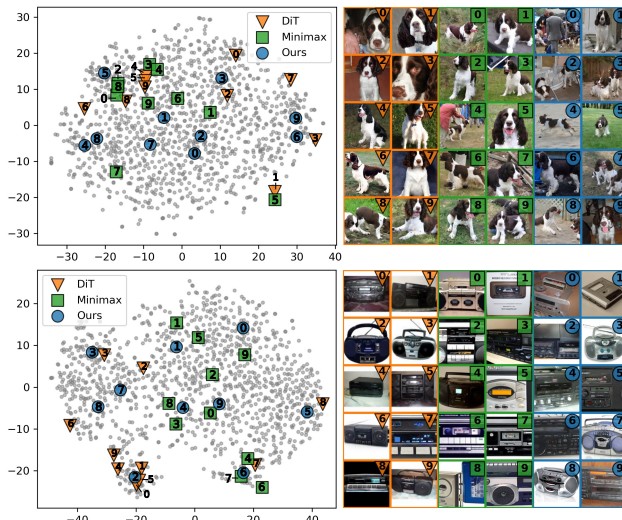

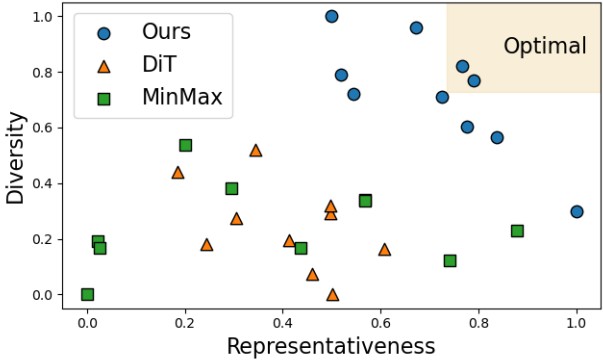

*Figure 5.* T-SNE plot showing the original samples (⬤) and the synthetic samples generated by different diffusion-based methods for two classes (English springer and cassette player) from ImageNet-1k. This visualization shows that DiT (Peebles & Xie, 2023) has limited diversity, Minmax (Gu et al., 2024) diffusion shows diversity but lacks full coverage, while our approach demonstrates mode diversity, achieving higher coverage.

*Figure 6.* Representative score versus Diversity score for each class on Nette for IPC 10 versus various models.

**Mode Guidance with DDIM.** Our approach, similar to classifier guidance (Nichol & Dhariwal, 2021), can be incorporated into DDIM using Algorithm 1. In Table 6, we compare the effect of our approach in DDPM and DDIM across LDM and DiT diffusion architectures. Our results demonstrate the effectiveness of our method with denoising samplers in both architectures, showcasing its flexibility with respect to diffusion architecture and sampler choice. This highlights the significant impact of our approach in enhancing the performance while being adaptable with different denoising diffusion models.

**Algorithm 1** Mode Guidance with DDIM sampling, given a diffusion model $\epsilon_\theta(x_t)$, an estimated mode $m_k$ and mode guidance scale $\lambda$.

---

Input: estimated mode $m_k$ and mode guidance scale $\lambda$
$x_T \leftarrow$ sample from $\mathcal{N}(0, \mathbf{I})$
**for all** $t$ from $T$ to 1 **do**
    $\mathbf{g_t} = (m_i - \hat{x}_0^t)$
    $\hat{\epsilon} \leftarrow \epsilon_\theta(x_t) - \sqrt{1 - \bar{\alpha}_t} \cdot \lambda \cdot \mathbf{g_t}$
    $x_{t-1} \leftarrow \sqrt{\bar{\alpha}_{t-1}} \left( \frac{x_t - \sqrt{1 - \bar{\alpha}_t}\hat{\epsilon}}{\sqrt{\bar{\alpha}_t}} \right) + \sqrt{1 - \bar{\alpha}_{t-1}}\hat{\epsilon}$
**end for**
**return:** $x_0$

---

*Table 6.* Performance comparison of diffusion models (LDM, DiT) with and without our approach, evaluated using DDPM and DDIM sampling methods on the Nette dataset on the IPC-10.

| Method | DDPM | DDIM |
|---|---|---|
| LDM | $60.3_{\pm 3.6}$ | $60.4_{\pm 3.1}$ |
| LDM + MGD$^3$ (Ours) | $61.9_{\pm 4.1}$ | $62.3_{\pm 1.1}$ |
| DiT | $58.8_{\pm 2.1}$ | $61.4_{\pm 2.4}$ |
| DiT + MGD$^3$ (Ours) | $66.4_{\pm 2.4}$ | $66.6_{\pm 0.6}$ |

## 6. Conclusion

Dataset distillation is an important task of condensing information from large training sets. Despite several efforts, the distilled datasets have limited representativeness and diversity in their synthetic samples. Our proposed method, leveraging latent diffusion with mode guidance, addresses this limitation and achieves state-of-the-art performance in dataset distillation across multiple benchmarks and experimental setups. Notably, our approach outperforms previous methods without requiring fine-tuning, as demonstrated by our results on ImageNette, ImageIDC, ImageNet-100, and ImageNet-1K. We conducted a detailed analysis of our method's key components and demonstrated their utility through rigorous ablation studies. Furthermore, we showed that our approach is compatible with general diffusion models, such as Text-to-Image Stable Diffusion, even when the training data does not overlap with the target dataset.

## Impact Statement

Efficient dataset distillation reduces storage and computational costs while maintaining high model performance. MGD$^3$ improves both accuracy and efficiency compared to prior methods, enabling large-scale dataset distillation with minimal performance trade-offs. This has significant implications for deep learning in resource-constrained environments, such as mobile AI and federated learning. By enhancing scalability, our work enables more effective model training with limited data while preserving diversity and representativeness.

## Acknowledgements

The authors thank the anonymous reviewers for their valuable feedback on earlier versions of this manuscript. This work was partially supported by a research gift from Cisco.

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

## A. When should guidance stop?

To determine when to stop the guidance, we assess mode guidance with $t_{stop}$ ranging from 50 to 0 in increments of 5 steps. A stop guidance of $t_{stop} = 50$ means no guidance, while $t_{stop} = 0$ means full guidance. Figure 7a shows that the optimal range to stop the guidance is between $t_{stop} = 30$ and $t_{stop} = 10$, with the peak at $t_{stop} = 20$. Additionally, Figure 7b illustrates that the guidance introduces more variability in the generation, with a more diverse set of backgrounds and poses. However, when the mode guidance is extended (e.g. $t_{stop} = 0$), it does not guarantee class consistency, as demonstrated in Figure 7b. This makes $t_{stop} = 25$ a good balance between generating a diverse set of backgrounds and maintaining class consistency.

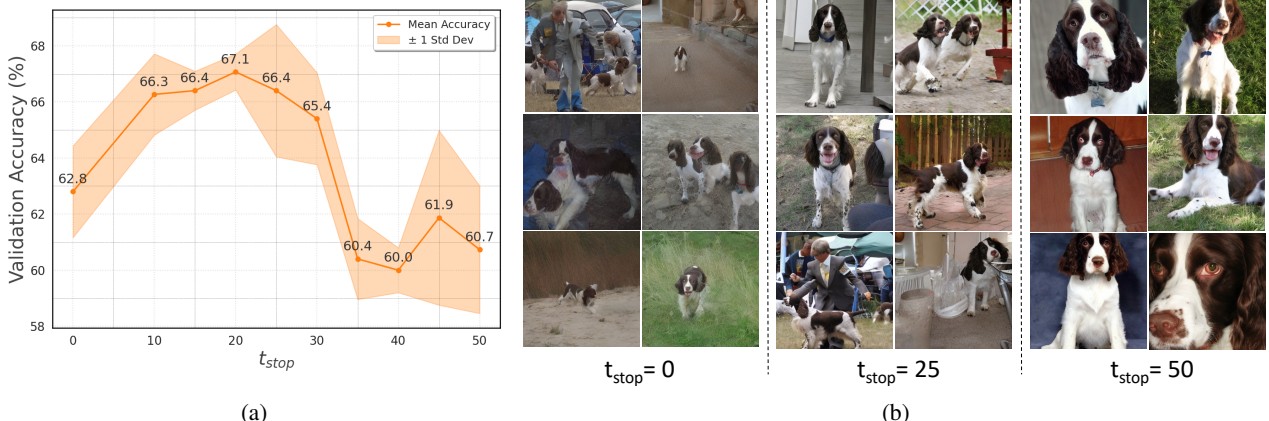

(a)          (b)

*Figure 7.* Ablation of the effect of $t_{stop}$, where $t_{stop} = 0$ denotes full guidance and $t_{stop} = 50$ denotes no guidance. (a) Shows validation accuracy versus $t_{stop}$ on ImageNette dataset. Best performance is achieved when $t_{stop}$ ranges between 20 and 30. (b) Shows generated images for the 'English Springer' class with full guidance ($t_{stop} = 0$), with early-stop guidance $t_{stop} = 25$ and no guidance ($t_{stop} = 50$). With early-stop guidance, the generated samples have more diversity w.r.t to the pose and background.

## B. Class wise Diversity and Representativeness

Figure 8 shows the diversity and representativeness of each distilled sample for ten classes in the ImageNet-1k dataset for DiT, MinMax, and ours. This Figure shows that our method is consistently have higher representativeness across all the classes in comparison to the previous methods. Overall, our method maintains high diversity across most of the samples within a class. We observe that both MinMax and DiT consistently have a few samples with very low diversity.

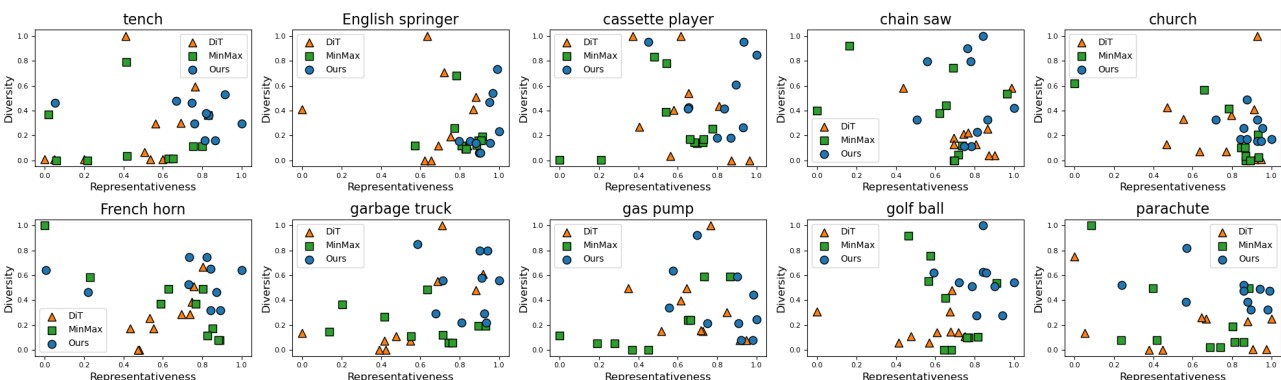

*Figure 8.* Representativeness versus Diversity by class for the distilled dataset from diffusion-based methods on 10 IPC of ImageNet-1k. Each point represents an image of the distilled dataset. DiT shows high representativeness but lacks diversity; MinMax shows diversity but lacks representativeness; Ours method shows both diversity and representativeness.

## C. Effect of Stop Guidance in Diversity and Representativeness

In order to understand how the stop guidance affects the diversity and representation of the distilled dataset, we perform an evaluation of these metrics on the ImageNette dataset for IPC 10 for various $t_{stop}$ ranging from 50 to 0; our results are shown in Figure 9. Our results show that applying mode guidance at any of the evaluated $t_{stop}$ values increases diversity, with the gains beginning to saturate beyond $t_{stop} = 30$. Surprisingly, delaying the stop guidance further into the reverse process (e.g $t_{stop} = 25$) leads to a noticeable increase in representativeness, while maintaining high diversity.

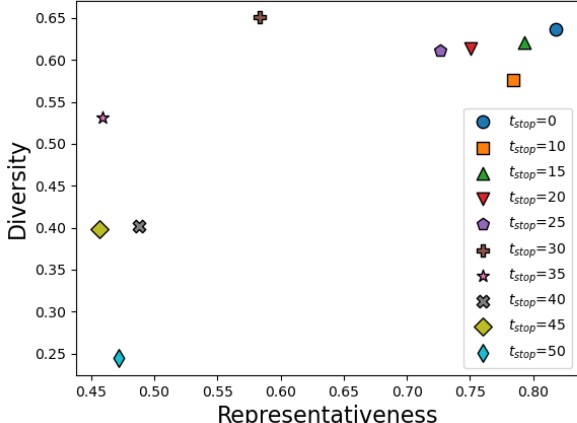

*Figure 9.* Representativeness versus Diversity versus $t_{stop}$. Each point represents a distilled dataset. Diversity and representativeness are obtained by computing the mean across all the samples in the distilled dataset. Stopping the mode guidance early in the reverse process ($t_{stop} = 45$ to $t_{stop} = 35$) promotes diversity. While prolonging the mode guidance between $t_{stop} = 35$ and $t_{stop} = 0$ increases representativeness.

## D. Effect of mode discovery algorithm

To investigate the impact of the mode discovery algorithm, we assess several strategies: random selection from the original dataset, $k$-means centroids, closest sample to $k$-means centroid, DBSCAN, spectral clustering, and Gaussian Mixture Models (GMM). The evaluation is conducted on ImageNette with IPC 10. For DBSCAN and spectral clustering, we compute the mean of each discovered cluster to represent a mode. For GMM, we use the mean of each Gaussian component. Mode guidance was applied with $t_{stop} = 25$ using the estimated modes from each method. The results, summarized in Table 7, show that GMM achieved the highest accuracy, slightly outperforming $k$-means centroids and other mode discovery techniques.

*Table 7.* Mode discovery algorithm versus Accuracy on ImageNette with IPC-10.

| Mode Discovery method | Accuracy |
|---|---|
| Random | $59.6_{\pm 1.8}$ |
| DBSCAN | $61.3_{\pm 1.9}$ |
| Spectral Clustering | $64.5_{\pm 2.1}$ |
| **GMM** | $\mathbf{66.9}_{\pm 0.4}$ |
| $k$-Means (closest sample) | $64.6_{\pm 0.4}$ |
| $k$-Means (centroid) | $66.4_{\pm 2.4}$ |

## E. Evaluation on ImageWoof

**ImageWoof.** We compare our method with SOTA in ImageWoof on IPC 10, 20, 50, 70, and 100 on various target architectures, as shown in Table 8. It is worth noticing that this dataset is a fine-grained dataset where all classes belong to

*Table 8.* Performance comparison with pre-trained diffusion models and other state-of-the-art methods on ImageWoof. All the results are reproduced by us for the 256×256 resolution. The missing results are due to out-of-memory. The best results are marked as **bold**. Higher is better.Results shown for the previous works are from (Gu et al., 2024).

| IPC (Ratio) | Test Model | Random | Herding (Welling, 2009) | DiT (Peebles & Xie, 2023) | DM (Zhao & Bilen, 2023) | IDC-1 (Kim et al., 2022) | GLaD (Cazenavette et al., 2023) | MinMaxDiff (Gu et al., 2024) | $MGD^3$ (**Ours**) | Full |
|---|---|---|---|---|---|---|---|---|---|---|
| 10 (0.8%) | ConvNet-6 | $24.3_{\pm1.1}$ | $26.7_{\pm0.5}$ | $34.2_{\pm1.1}$ | $26.9_{\pm1.2}$ | $33.3_{\pm1.1}$ | $33.8_{\pm0.9}$ | $\mathbf{37.0}_{\pm1.0}$ | $34.73_{\pm1.1}$ | $86.4_{\pm0.2}$ |
| | ResNetAP-10 | $29.4_{\pm0.8}$ | $32.0_{\pm0.3}$ | $34.7_{\pm0.5}$ | $30.3_{\pm1.2}$ | $39.1_{\pm0.5}$ | $32.9_{\pm0.9}$ | $39.2_{\pm1.3}$ | $\mathbf{40.4}_{\pm1.9}$ | $87.5_{\pm0.5}$ |
| | ResNet-18 | $27.7_{\pm0.9}$ | $30.2_{\pm1.2}$ | $34.7_{\pm0.4}$ | $33.4_{\pm0.7}$ | $37.3_{\pm0.2}$ | $31.7_{\pm0.8}$ | $37.6_{\pm0.9}$ | $\mathbf{38.5}_{\pm2.5}$ | $89.3_{\pm1.2}$ |
| 20 (1.6%) | ConvNet-6 | $29.1_{\pm0.7}$ | $29.5_{\pm0.3}$ | $36.1_{\pm0.8}$ | $29.9_{\pm1.0}$ | $35.5_{\pm0.8}$ | - | $37.6_{\pm0.2}$ | $\mathbf{39.0}_{\pm3.46}$ | $86.4_{\pm0.2}$ |
| | ResNetAP-10 | $32.7_{\pm0.4}$ | $34.9_{\pm0.1}$ | $41.1_{\pm0.8}$ | $35.2_{\pm0.6}$ | $43.4_{\pm0.3}$ | - | $\mathbf{45.8}_{\pm0.5}$ | $43.6_{\pm1.6}$ | $87.5_{\pm0.5}$ |
| | ResNet-18 | $29.7_{\pm0.5}$ | $32.2_{\pm0.6}$ | $40.5_{\pm0.5}$ | $29.8_{\pm1.7}$ | $38.6_{\pm0.2}$ | - | $\mathbf{42.5}_{\pm0.6}$ | $41.9_{\pm2.1}$ | $89.3_{\pm1.2}$ |
| 50 (3.8%) | ConvNet-6 | $41.3_{\pm0.6}$ | $40.3_{\pm0.7}$ | $46.5_{\pm0.8}$ | $44.4_{\pm1.0}$ | $43.9_{\pm1.2}$ | - | $53.9_{\pm0.6}$ | $\mathbf{54.5}_{\pm1.6}$ | $86.4_{\pm0.2}$ |
| | ResNetAP-10 | $47.2_{\pm1.3}$ | $49.1_{\pm0.7}$ | $49.3_{\pm0.2}$ | $47.1_{\pm1.1}$ | $48.3_{\pm1.0}$ | - | $56.3_{\pm1.0}$ | $\mathbf{56.5}_{\pm1.9}$ | $87.5_{\pm0.5}$ |
| | ResNet-18 | $47.9_{\pm1.8}$ | $48.3_{\pm1.2}$ | $50.1_{\pm0.5}$ | $46.2_{\pm0.6}$ | $48.3_{\pm0.8}$ | - | $57.1_{\pm0.6}$ | $\mathbf{58.3}_{\pm1.4}$ | $89.3_{\pm1.2}$ |
| 70 (5.4%) | ConvNet-6 | $46.3_{\pm0.6}$ | $46.2_{\pm0.6}$ | $50.1_{\pm1.2}$ | $47.5_{\pm0.8}$ | $48.9_{\pm0.7}$ | - | $\mathbf{55.7}_{\pm0.9}$ | $55.1_{\pm2.5}$ | $86.4_{\pm0.2}$ |
| | ResNetAP-10 | $50.8_{\pm0.6}$ | $53.4_{\pm1.4}$ | $54.3_{\pm0.9}$ | $51.7_{\pm0.8}$ | $52.8_{\pm1.8}$ | - | $58.3_{\pm0.2}$ | $\mathbf{60.2}_{\pm2.4}$ | $87.5_{\pm0.5}$ |
| | ResNet-18 | $52.1_{\pm1.0}$ | $49.7_{\pm0.8}$ | $51.5_{\pm1.0}$ | $51.9_{\pm0.8}$ | $51.1_{\pm1.7}$ | - | $58.8_{\pm0.7}$ | $\mathbf{59.7}_{\pm2.7}$ | $89.3_{\pm1.2}$ |
| 100 (7.7%) | ConvNet-6 | $52.2_{\pm0.4}$ | $54.4_{\pm1.1}$ | $53.4_{\pm0.3}$ | $55.0_{\pm1.3}$ | $53.2_{\pm0.9}$ | - | $\mathbf{61.1}_{\pm0.7}$ | $60.1_{\pm1.2}$ | $86.4_{\pm0.2}$ |
| | ResNetAP-10 | $59.4_{\pm1.0}$ | $61.7_{\pm0.9}$ | $58.3_{\pm0.8}$ | $56.4_{\pm0.8}$ | $56.1_{\pm0.9}$ | - | $64.5_{\pm0.2}$ | $\mathbf{66.5}_{\pm1.0}$ | $87.5_{\pm0.5}$ |
| | ResNet-18 | $61.5_{\pm1.3}$ | $59.3_{\pm0.7}$ | $58.9_{\pm1.3}$ | $60.2_{\pm1.0}$ | $58.3_{\pm1.2}$ | - | $65.7_{\pm0.4}$ | $\mathbf{68.8}_{\pm0.7}$ | $89.3_{\pm1.2}$ |

dog breeds. Due to its granularity of features, we trained DiT XL/2 on the ImageWoof dataset with just the simple loss mentioned in Eq. 2 following the same training epochs as (Gu et al., 2024). Our method outperforms the previous SOTA across various IPC values for different target architectures. Notably, our method demonstrates superior performance in all IPC values for the ResNet-18 architecture, achieves SOTA in IPC 10, 50, 70, and 100 with the ResNetAP-10 architecture, and delivers the best performance in IPC 20 and 50 with the ConvNet-6 architecture.

## F. Effect of mode guidance scale $\lambda$

To study how the mode guidance scale $\lambda$ affects performance, we evaluate the various values for $\lambda$ on ImageNette with IPC 10 with ResNetAP-10. Our results show that when the mode guidance is too high, it's catastrophic for the distilled data, dropping the performance significantly; however, the best parameter is achieved by $\lambda = 0.1$.

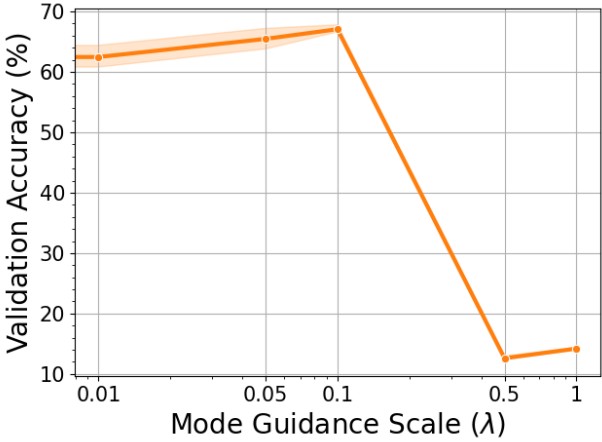

*Figure 10.* Effect of guidance scale on performance.

## G. Diversity Class-Wise diversity score

We calculate the diversity score for each class by averaging the diversity score across all the samples. Table 9 shows the diversity score for each class for DiT, MinMax, and Mode Guidance. Our method consistently generates a more diverse set for each class for ImageNette than the other methods.

*Table 9.* Results: Comparison of per-class diversity scores on ImageNette with IPC-10

| class | DiT | MinMax | Ours |
|---|---|---|---|
| tench | 0.35 | 0.18 | **0.82** |
| English springer | 0.65 | 0.33 | **0.62** |
| cassette player | 0.55 | 0.52 | **1.00** |
| chain saw | 0.00 | 0.37 | **0.55** |
| church | 0.54 | 0.41 | **0.77** |
| French horn | 0.21 | 0.13 | **0.54** |
| garbage truck | 0.44 | 0.38 | **0.76** |
| gas pump | 0.50 | 0.24 | **0.67** |
| golf ball | 0.20 | 0.33 | **0.78** |
| parachute | 0.08 | 0.48 | **0.79** |
| Average | 0.35 | 0.34 | **0.73** |

## H. Hard-Label versus Soft-label Protocols

We conduct further analysis on ImageNet-100, where we test our approach from IPC-10 up to IPC-100. As illustrated in Table 10, our performance steadily improves, reaching $57.8_{\pm0.2}$ with the hard-label protocol. Additionally, we compare the performance of ImageNet-100 using soft-label training on IPC-10, 20, 50, and 100. The results underscore a substantial performance boost when employing soft-labels.

*Table 10.* Evaluation of training with hard-labels versus soft labels in ImageNet-100 training with ResNet18.

| Method | Labels | IPC10 | IPC20 | IPC50 | IPC 100 |
|---|---|---|---|---|---|
| **MGD$^3$ (Ours)** | Hard-Label | $23.6_{\pm0.4}$ | $32.6_{\pm0.4}$ | $51.8_{\pm0.2}$ | $57.8_{\pm0.2}$ |
| | Soft-label | $34.0_{\pm1.0}$ | $50.2_{\pm0.7}$ | $69.2_{\pm0.4}$ | $75.8_{\pm0.3}$ |

## I. Evaluation Technical Details

For the hard-label protocol, we follow the evaluation method described in (Gu et al., 2024). We train our model on a synthetic dataset for 1500 epochs for IPC values of 20, 50, and 100, and extend the training to 2000 epochs for an IPC value of 10. We use Stochastic Gradient Descent (SGD) as the optimizer, setting the learning rate at $0.01$. We use a learning rate decay scheduler at the $2/3$ and $5/6$ points of the training process, with the decay factor (gamma) set to $0.2$. Cross-entropy was used as the Loss objective.

For the soft-label protocol, we follow the evaluation used by (Gu et al., 2024; Sun et al., 2024) for ImageNet-1k evaluation. We evaluate the model by training a network for 300 epochs with Resnet-18 architecture as both teacher and student. We use the AdamW optimizer, with a learning rate set at $0.001$, a weight decay of $0.01$, and the parameters $\beta_1 = 0.9$ and $\beta_2 = 0.999$.

## J. Visualization of Denoising Trajectories with Mode Guidance for Different $t_{stop}$

Figure 11 illustrates the effect of Stop Guidance ($t_{stop}$) on the generated image. Stopping early (e.g., $t_{stop} = 45$) can introduce features unrelated to the target class, such as the baby face in the top row. Conversely, extending guidance too long (e.g., $t_{stop} = 0$) degrades image quality.

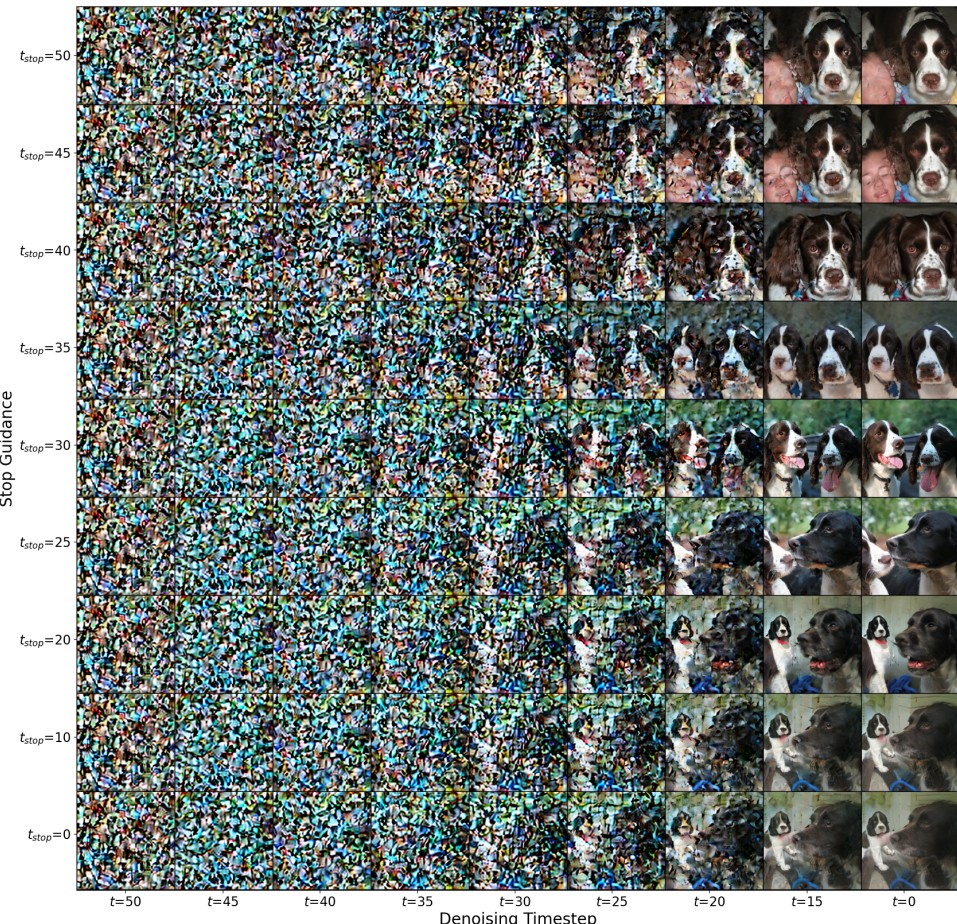

*Figure 11.* Generated images through the denoising process for different values of $t_{stop}$.

