# OpenReview forum: "MGD$^3$ : Mode-Guided Dataset Distillation using Diffusion Models"
_ICML.cc/2025/Conference — ICML 2025 oral_

### Official Review · Reviewer_y4kY · 2025-02-24

**Overall Recommendation:** 4

**Summary:**

The paper proposes a mode-guided diffusion model for dataset distillation. It aims to address the limitations of existing dataset distillation methods, such as insufficient sample diversity and high computational costs. The key idea is to leverage a pre-trained diffusion model without fine-tuning using distillation losses. It has three stages: Mode Discovery to identify data modes, Mode Guidance to enhance intra-class diversity, and Stop Guidance to improve the quality of synthetic samples. Experimental results on multiple datasets show that the method outperforms existing approaches.

## update after rebuttal
I agree to accept the paper.

**Claims And Evidence:**

Yes.

**Essential References Not Discussed:**

No obvious essential missing reference to my best knowledge.

**Experimental Designs Or Analyses:**

The experimental designs are relatively sound. The authors use appropriate baselines, different evaluation protocols, and conduct ablation studies. But the number of runs in the comparison with sota methods could be increased to obtain more stable results.

**Methods And Evaluation Criteria:**

The proposed methods are well-suited for the problem of dataset distillation. And the evaluation criteria, including accuracy and diversity metrics, are appropriate for assessing the performance of dataset distillation methods.

**Other Comments Or Suggestions:**

Typos: Unified 'dataset' and 'data set' in the Introduction section.

**Other Strengths And Weaknesses:**

Strengths:
The proposed MGD3 method is novel in its approach to dataset distillation. The three-stage process of Mode Discovery, Mode Guidance, and Stop Guidance is a creative combination of existing concepts in diffusion models and dataset distillation, providing a unique solution to the problem of sample diversity and representativeness.
And the paper is generally in good clarity. For instance, Figure 1 clearly differentiates optimization-based and generative-based dataset distillation.

Weaknesses:
Some aspects of the method, like using clustering for mode discovery, may not be entirely new in the context of generative models. The novelty might be diluted when compared to truly revolutionary ideas in the field.
And the theoretical depth of the method could be further strengthened. There is few original formula to explain why the three-stage process works optimally in terms of diversity and representativeness.

**Questions For Authors:**

Question 1: In the mode discovery stage, the authors use K-Means centroids. Have you considered other more advanced clustering algorithms, and what are the potential advantages or disadvantages? If other algorithms can improve the performance, it might further strengthen your method.
Question 2: The performance improvement of your method is demonstrated on several Image datasets. But how about the performance on datasets with more complex distributions or smaller sample sizes? A response showing good performance on such datasets would enhance the generality of your method.

**Relation To Broader Scientific Literature:**

Compared with prior works that rely on fine-tuning with distillation losses, this paper's training-free approach is a new contribution in this field.

**Theoretical Claims:**

The paper does not have complex theoretical proofs. But the concepts and operations in the method, such as mode discovery using clustering algorithms and mode guidance based on the diffusion process, are based on established theories in machine learning and diffusion models. There are no obvious theoretical flaws in the presented ideas.

---

> ### Author Rebuttal · Authors · 2025-04-01
>
> We appreciate the reviewer's positive assessment of our work, particularly our novel three-stage mode-guided approach, clear presentation, and contribution as a training-free dataset distillation method.
>
> ## Addressing Concerns About Our Novelty
> In dataset distillation, modes have been used to enhance diversity or sample from prototypes of the dataset distribution. Given these dataset prototypes, the goal is to generate a small dataset that maintains a similar distribution as the original dataset. Prior work [1] introduced noisy modes as $x_0$ instead of random noise; however, the high noise levels in the initial denoising steps often lead to outputs that are neither diverse nor fully representative. This explains why Disentangled Diffusion performs similarly to, or even worse than, simply using a pretrained diffusion model. The novelty of our approach lies in its efficient technical implementation, which requires no additional training and generalizes across various datasets and diffusion architectures, including Latent Diffusion (Class-to-Image), DiT, and Stable Diffusion (Text-to-Image), which is not the case of D$^4$M as shown in our experiments (Table 1 and Figure 4c).
>
> [1] Su, Duo, et al. "D^ 4: Dataset Distillation via Disentangled Diffusion Model." CVPR . 2024.
>
> ## Mode Guidance Theoretical Support
>
> We aim to generate samples close to a given mode $m_i$ using mode-classifier guidance:
>
> $$\nabla_{x_t} \log p_t(x_t | m_i) = \nabla_{x_t} \log p_t(x_t) + \nabla_{x_t} \log p_t(m_i | x_t)$$
>
> Defining mode-classifier probability (how far is the current noise from the mode $m_i$) as:
>
> $$p(m_i | x_t) \propto \exp\left(-\frac{1}{2} |x_t - m_i|^2\right)$$
>
> Taking the gradient and introducing guidance scale $\lambda$:
>
> $$\nabla_{x_t} \log p_t(m_i | x_t) \approx -\lambda (x_t - m_i)$$
>
> Substituting into the classifier guidance equation:
>
> $$\nabla_{x_t} \log p_t(x_t | m_i) = \nabla_{x_t} \log p_t(x_t) - \lambda (x_t - m_i)$$
>
> With the score function related to noise prediction:
>
> $$\nabla_{x_t} \log p_t(x_t) \approx -\frac{\tilde{\boldsymbol{\epsilon}}_\theta(x_t, t)}{\sigma_t}$$
>
> Leading to modified noise prediction:
>
> $$\epsilon_\theta(x_t, t) = \tilde{\boldsymbol{\epsilon}}_\theta(x_t, t) + \lambda \sigma_t (m_i - x_t)$$
>
> Similar to Classifier Guidance, the model pushes toward images near mode $m_i$, with $\lambda$ controlling guidance strength.
>
> ## Experiments with Other Mode Discovery Techniques
> We tested more advanced mode discovery algorithms as suggested. GMM and Spectral Clustering improved our method in some settings:
>
> |Method|IPC 10|IPC 20|IPC 50|
> |---|---|---|---|
> |GMM|66.9±0.6|70.8±2.9|79.7±1.7|
> |Spectral Clustering|64.5±2.1|70.5±2.0|79.7±2.0|
> |$k$-means|66.4±2.4|71.2±0.5|79.5±1.3|
>
> Table 1: Experiments with suggested mode discovery methods on ImageNette.
>
> ## Experiment with Complex distributions (imbalanced dataset)
> We tested our method on class-imbalanced ImageNet-100 subsets. We created imbalanced datasets with "low sample" classes with 10-20 images and "normal" classes with  800 images. We vary low sample classes from 10 to 100 to generate a dataset with different degrees of imbalance and evaluate our method on IPC 10. The original test dataset is used to compute the accuracy. We reported the performance choosing 10 samples per class randomly of the imbalanced dataset,
>
> | Low samples classes | Random     | Ours       |
> | ------------ | ---------- | ---------- |
> | 0| 19.1 ± 0.4 | 25.8 ± 0.5 |
> | 10| 19.3± 0.2  | 25.1 ± 1.0 |
> | 20 | 18.2± 0.7  | 24.4 ± 0.2 |
> | 30| 18.8 ± 0.3 | 24.0 ± 0.2 |
> | 40 | 17.6 ± 0.1 | 23.9 ± 0.5 |
> | 50 | 17.7± 0.4  | 23.8 ± 0.1 |
> | 60  | 18.9± 0.5  | 23.4 ± 0.5 |
> | 70| 19.2± 0.2  | 23.0 ± 0.8 |
> | 80 | 19.0± 0.1  | 23.0 ± 1.0 |
> | 90 | 19.3± 0.2  | 22.2 ± 0.3 |
> | 100 | 18.3± 0.5  | 20.8 ± 0.2 |
>
> Even when estimating modes from complex distributions with limited samples, our method maintains strong performance, consistently outperforming random selection. Notably, using a DiT trained on a balanced dataset achieves an accuracy of 23.3 ± 0.4, which our method surpasses even when 60% of the classes have low sample sizes.
>
> ## Measuring Diversity and Representativeness Across MGD³ Components
> Our three-stage process optimizes diversity and representativeness while maintaining realism. We include classification accuracy from an ImageNet-trained model to assess realism:
>
> | Method         | Diversity | Representativeness | Classification Acc |
> | -------------- | --------- | ------------------ | ------------------ |
> | DiT| 0.28| 0.55| 0.99|
> | Mode Discovery |0.65| 0.24| 0.91|
> | MG| 0.71| 0.82| 0.83|
> | MG+StopG | 0.72| 0.80| 0.95|
>
> Mode Guidance improves diversity (0.71) and representativeness (0.82) but lowers accuracy (0.83), indicating reduced realism. Stop Guidance mitigates this, restoring accuracy (0.95) while preserving diversity (0.72) and representativeness (0.80). This achieves better trade-off than DiT (0.99 accuracy but significantly lower diversity and representativeness).

---

> > ### Comment · Reviewer_y4kY · 2025-04-03
> >
> > After carefully reviewing the authors' response, I found their clarifications satisfactory and have consequently elevated my rating to 4.

---

> > > ### Author Response · Authors · 2025-04-08
> > >
> > > Dear Reviewer y4kY,
> > >
> > > Thank you for your thoughtful and constructive review. We sincerely appreciate the time you took to reassess our work and are pleased that we were able to address your concerns.

---

### Official Review · Reviewer_Zv9A · 2025-03-01

**Overall Recommendation:** 4

**Summary:**

This paper focuses on the dataset distillation task based on generative models. Specifically, the authors argue that the current methods cannot guarantee the sample diversity, which is essential for model training. Based on this observation, the authors propose a mode-guided diffusion model. Distinct data modes are first discovered from the original data distribution. Then the modes are used to guide the generation process to enhance intra-class diversity. The proposed method surpasses previous state-of-the-art methods by at least 1.6%, with much less computational cost required.
## update after rebuttal
Thanks for the author's reply. I would keep my recommendation of acceptance.

**Claims And Evidence:**

The claimed contributions are well supported by the numerical results. The proposed method achieves state-of-the-art validation performance on the full ImageNet-1K. And the improved diversity and representativeness are supported by designed metrics.

**Essential References Not Discussed:**

N/A

**Experimental Designs Or Analyses:**

1. The proposed method guides the image generation with discovered modes. And the modes are actually latent embeddings. It would be interesting to provide an experimental comparison with directly using these samples for training under the same IPC. I am curious how the diffusion model helps with the generation quality if there already exist several selected modes.
2. It is not sufficiently clear how the experiments are carried out and how the comparisons are conducted with the baseline methods. Are the authors building their method on top of the baselines? Or, are they simply methods that the authors used to compare with?

**Methods And Evaluation Criteria:**

1. The discovered modes well cover the original data distribution. However, is there the possibility that the diversity of the generated samples is in turn restricted by the modes as there are only a limited number of them?

**Other Comments Or Suggestions:**

N/A

**Other Strengths And Weaknesses:**

Strengths:

**Weaknesses:**
1. It is not clear for each sample during generation, which mode is selected at the inference stage.

**Questions For Authors:**

I have two main questions:
1. How does the diffusion process help improve the data quality compared with the selected modes?
2. Is there a possibility that the discovered modes restrict the diversity of the generated data with a large IPC? How do the authors deal with this condition?

**Relation To Broader Scientific Literature:**

Methods based on generative models are a new trend in the field of dataset distillation. The proposed method is a useful tool at inference time to generate diverse samples covering the original distribution. The method can be easily plugged into any diffusion models.

**Theoretical Claims:**

There is no explicit theoretical analysis presented.

---

> ### Author Rebuttal · Authors · 2025-04-01
>
> We appreciate the reviewer’s positive assessment of our work, particularly the recognition of our method’s state-of-the-art performance, its effectiveness in improving sample diversity, and its relevance to generative model-based dataset distillation. We also sincerely appreciate the reviewer’s positive evaluation of our claims and supporting evidence, which reinforces the validity and impact of our contributions. We address the reviewer’s insightful questions regarding the positive impact of the diffusion process on data quality and the question about mode selection in our response below.
>
> ## Clarifying Mode Sampling During Inference:
>
> Given a IPC = N, our method find N modes then we sample 1 image per mode.
>
> ## Experiment using Latent Modes as Samples
>
> As suggested by the reviewer, we tested using mode images directly as samples on ImageNette (IPC 10). Results in Table 1 show this performs poorly—worse than using the closest real sample to the mode. This occurs because the latent mode is an average of the cluster, creating a blurry image. MGD³ improves performance by using diffusion to generate real images near the mode rather than using the mode directly.
>
> | Method | IPC 10 |
> | -------------- | -------------- |
> | Closest Sample | 57.1 ± 1.3 |
> | Latent Mode | 37.7 ± 1.1 |
> | MGD$^3$ | **66.4 ± 2.4** |
>
> Table 1: Results for suggested experiment with latent modes.
>
> ## Clarification about comparisons
> Our method was implemented independently, not built upon baselines, and evaluated under identical conditions as existing approaches. We compared our approach with pretrained diffusion models (LDM, DiT) and incorporated these alongside Disentangled Diffusion for comprehensive evaluation. In text-to-image setups, we benchmarked against Stable Diffusion and D$^4$M. Notably, our method outperformed our closest competitor MinMaxDiffusion even without requiring the finetuning that MinMaxDiffusion needs for dataset distillation.
>
> ## Addressing Mode Limitation Concerns in High-IPC Settings
> When the selected IPC is larger than the true number of modes in the data for a class, our method captures fine-grained variations within each mode without affecting the diversity of the generated samples. This is controlled by the mode discovery algorithm, and we observe that our default k-means approach works well in practice.

---

> > ### Comment · Reviewer_Zv9A · 2025-04-02
> >
> > Thanks for the reply. Some of my questions are not fully addressed. The authors claim that for high IPC, their method captures fine-grained variations without affecting the diversity. However, there is no evidence of it. Given the sufficient space from the 5,000 character limit, the authors are expected to give a more detailed explanation or evidence.
> >
> > In line 312 left column of the manuscript, the authors first state that they adopt two baselines: DiT and MinMax diffusion. Then there are DiT and LDM actually serving as baselines in result tables. It is misleading, while the authors seem not to have noticed it.
> >
> > Given the technical solidness of this paper, I would like to currently keep my initial score of acceptance. However, please do make clarifications about the raised questions for the manuscript.

---

> > > ### Author Response · Authors · 2025-04-03
> > >
> > > We appreciate the reviewer’s thoughtful feedback and the opportunity to clarify our response.
> > >
> > > Diversity in generated samples can be assessed from two perspectives: **(1) mode coverage**, ensuring that all underlying modes in the data distribution are well-represented, and **(2) pairwise distance**, which measures how spread out the samples are in the data space. Our method focuses on maintaining mode coverage by capturing the estimated modes and effectively distributing samples across different underlying structures in the data. However, as IPC increases beyond the true number of modes, samples become more densely packed within each mode, leading to a natural reduction in diversity in terms of pairwise distance. This effect can be observed in the table below:
> > >
> > > | IPC | Diversity |
> > > | --- | --------- |
> > > | 10  | 0.68      |
> > > | 20  | 0.28      |
> > > | 40  | 0.11      |
> > > | 50  | 0.07      |
> > >
> > > While a higher IPC brings samples closer together, the benefits of additional samples enable a more refined approximation of the distribution.
> > >
> > > Additionally, we acknowledge the inconsistency in the baseline descriptions. To clarify: LDM and DiT serve as baselines representing pre-trained diffusion models, while MinMaxDiffusion is included to compare against a diffusion model fine-tuned on dataset distillation. We will make this distinction clearer in the manuscript to avoid any ambiguity.
> > >
> > > Thank you again for your careful review and constructive suggestions.

---

### Official Review · Reviewer_24Ct · 2025-03-12

**Overall Recommendation:** 4

**Summary:**

This paper presents Mode-Guided Dataset Distillation using Diffusion Models (MGD³), a novel dataset distillation method that improves diversity and representativeness in distilled datasets without requiring additional fine-tuning of diffusion models. The approach comprises three key components: Mode Discovery to identify distinct data modes within a class to enhance sample diversity. Mode Guidance to guide the diffusion process toward specific modes to avoid redundant samples. Stop Guidance to halt guidance at a certain timestep to maintain sample fidelity.

## update after rebuttal
The authors adequately addressed my concerns, and I have raised my score to Accept.

**Claims And Evidence:**

1. The proposed method improves dataset diversity and representativeness without additional fine-tuning. This is demonstrated through quantitative comparisons with MinMax Diffusion and other SOTA methods across multiple datasets (Tables 1-3). The t-SNE visualizations (Figure 6) illustrate better mode coverage than previous methods. The diversity vs. representativeness plots (Figures 5, 8) further confirm these properties.

2. Mode Guidance effectively improves sample diversity while maintaining class consistency. Ablation studies (Table 4) indicate that each component (Mode Discovery, Mode Guidance, Stop Guidance) contributes incrementally to performance gains. Guidance stopping strategies (Figure 7) provide insights into optimal stopping points for balancing diversity and fidelity.

3. The proposed method is computationally efficient. Compared to MinMax Diffusion (10+ hours) and IDC-1 (100+ hours), MGD³ generates a dataset in 0.42 hours for ImageNet-100 (IPC-10), showcasing significant efficiency improvements.

4. The claim of "MGD³ consistently outperforms all existing methods." is problematic. While MGD³ achieves SOTA on ImageNette, ImageIDC, and ImageNet-1K, it does not fully surpass IDC-1 on ImageNet-100. The paper should clarify that MGD³ provides a trade-off between performance and efficiency, rather than always achieving the highest accuracy.

**Essential References Not Discussed:**

The paper lacks discussion of several key recent works on decoupled dataset distillation methods [1-4].

[1] Squeeze, Recover and Relabel: Dataset Condensation at ImageNet Scale From A New Perspective

[2] Generalized Large-Scale Data Condensation via Various Backbone and Statistical Matching

[3] On the Diversity and Realism of Distilled Dataset: An Efficient Dataset Distillation Paradigm

[4] Elucidating the Design Space of Dataset Condensation

**Experimental Designs Or Analyses:**

The experiments are well-designed, comparing MGD³ with strong SOTA baselines across multiple IPC values. The ablation study effectively isolates the impact of Mode Discovery, Mode Guidance, and Stop Guidance. The computational efficiency analysis strengthens the argument for real-world applicability. Sensitivity analysis of hyperparameters (λ, t_stop, number of modes) is only partially addressed (Table 5, Figure 10). Are these values consistent across datasets? The ImageWoof evaluation is informative but lacks discussion.

**Methods And Evaluation Criteria:**

The proposed methodology is well-structured and logically motivated. The benchmark datasets (ImageNette, ImageNet-100, ImageNet-1K) and evaluation metrics (classification accuracy under hard-label and soft-label protocols) align with the dataset distillation task. The use of Mode Discovery in the VAE latent space is novel and helps mitigate mode collapse. Comparisons with both diffusion-based and optimization-based methods strengthen the evaluation. Ablation studies systematically justify the effectiveness of each component.

**Other Comments Or Suggestions:**

This paper should include an impact statement.

**Other Strengths And Weaknesses:**

See above.

**Questions For Authors:**

1. Do the optimal values of λ and t_stop remain consistent across datasets (e.g., ImageNette vs. ImageNet-100 vs. ImageNet-1K)? Or do they need dataset-specific tuning?

2. Did you experiment with other mode discovery techniques (e.g., Gaussian Mixture Models (GMM), Spectral Clustering, or Density-Based Approaches) for estimating modes?

3. Testing on few-shot learning scenarios: Can MGD³ help in ultra-low IPC regimes (e.g., IPC = 1 or 5)?

**Relation To Broader Scientific Literature:**

Unlike optimization-based distillation, which requires repeated executions to synthesize datasets of different sizes, MGD³ leverages diffusion models to directly sample from a learned data distribution, making it more scalable.

**Theoretical Claims:**

The paper builds on diffusion model guidance principles (e.g., classifier-free guidance from Ho & Salimans, 2022). This derivation aligns with score-based diffusion models but lacks a formal proof of why this improves diversity without fine-tuning. A theoretical justification (e.g., a distributional analysis) could further support why Stop Guidance prevents over-conditioning.

---

> ### Author Rebuttal · Authors · 2025-04-01
>
> We appreciate the reviewer’s thoughtful feedback and recognition of our contributions. Specifically, we are pleased that the reviewer highlights the novelty of our Mode-Guided Dataset Distillation using Diffusion Models (MGD³), its computational efficiency, the well-structured methodology, and the extensive experimental evaluation, including ablation studies and comparisons with strong baselines.
>
> ## Clarifying Performance and Efficiency Trade-off
> We acknowledge that while MGD³ achieves state-of-the-art performance across multiple benchmarks, IDC-1 slightly outperforms it in specific cases on ImageNet-100 (e.g., ResNet-18 with 10 samples). However, MGD³ consistently provides strong accuracy while significantly improving computational efficiency, making it more scalable than IDC-1. MGD³ achieves the best performance in several configurations (e.g., ConvNet-6 and ResNetAP-10 with 20 samples). We'll revise our statement to explicitly highlight this trade-off between accuracy and efficiency.
>
> ## References
> Although we cited [1] and [3] in the main paper, we will expand our discussion to clarify how MGD³ relates to works [1-4], particularly regarding efficiency and scalability, and include comparisons in Table 1(reviewer rCkP).
>
> ## Impact Statement
> Efficient dataset distillation reduces storage and computational costs while maintaining high model performance. MGD³ improves both accuracy and efficiency compared to prior methods, enabling large-scale dataset distillation with minimal performance trade-offs. This has significant implications for deep learning in resource-constrained environments, such as mobile AI and federated learning. By enhancing scalability, our work enables more effective model training with limited data while preserving diversity and representativeness.
>
> The Impact Statement will be included in the final revision.
>
> ## Hyperparameter Experiments
> While our main experiments used fixed hyperparameters for consistency (still achieving SOTA results), we conducted additional experiments with varying hyperparameters for each dataset as requested. Results for different values of λ and t_stop on Nette, IDC, and ImageNet-100 with IPC 10 are shown below:
>
> | Dataset      | $\lambda$ = 0.01 | $\lambda$ = 0.05 | $\lambda$ = 0.07 | $\lambda$ = 0.1 | $\lambda$ = 0.3 | $\lambda$ = 0.5 |
> | ------------ | ---------------- | ---------------- | ---------------- | --------------- | --------------- | --------------- |
> | Nette        | 62.5±1.81        | 65.5±1.7         | 65.5±1.0         | **67.1±0.6**    | 43.4±2.1        | 12.6±0.4        |
> | IDC          | 48.4±1.31        | 54.9±1.1         | 53.3±0.7         | **55.9±2.1**    | 32.9±1.1        | 16.7±2.6        |
> | ImageNet-100 | 25.0±0.38        | 27.1±0.4         | **27.2±0.4**     | 25.77±0.5       | 12.0±0.4        | 2.3±0.1         |
> | Mean         | 45.3             | 49.2             | 48.7             | **49.6**        | 29.4            | 9.2             |
>
> **Table 2: Results with various mode guidance scale parameters (λ)**
>
> | Dataset      | $t_{stop}$ = 0 | $t_{stop}$ =10 | $t_{stop}$ = 20 | $t_{stop}$ = 25 | $t_{stop}$ = 30 | $t_{stop}$ = 40 | $t_{stop}$ = 50 |
> | ------------ | -------------- | -------------- | --------------- | --------------- | --------------- | --------------- | --------------- |
> | Nette | 62.8±1.6 | 66.3±1.5 | **67.1±0.6**| 66.4±2.4| 65.4±1.6 | 60.0±0.8| 60.7±2.3|
> | IDC | 52.8±4.2 | 53.5±2.0| 52.1±0.8| **55.9±2.1**| 53.2±2.6| 47.8±3.5| 42.5±1.3|
> | ImageNet-100 | 23.1±1.0| 25.1±0.4| **26.2±0.5**| 25.8±0.5| 24.1±0.5| 21.2±0.3| 23.3±0.3|
> | Mean         | 46.2           | 48.3           | 48.5            | **49.4**        | 47.6            | 43.0            | 42.2            |
>
> **Table 3: Results with various values for $t_{stop}$**
>
> While performance can be improved for specific datasets by adjusting parameters, our default parameters result in higher average scores across datasets.
>
> ## Discussion on ImageWoof
> We included a discussion of ImageWoof results in the appendix (line 677-679).
>
> ## Experiments with Other Mode Discovery Techniques
> Please refer to the Response of reviewer y4kY.
>
> ## Addressing Ultra-Low IPC Performance: MGD³'s Scaling Advantages
> The results from existing works on dataset distillation suggest that generative methods for dataset distillation underperform in low-IPC setups (e.g., IPC = 1) compared to optimization-based methods, which explicitly optimize the small dataset but at a higher computational cost. However, generative models scale better to higher IPC (e.g., IPC = 50) while maintaining lower computational cost, i.e., within a given time frame,  generative models can efficiently generate higher-IPC datasets compared to the optimization-based methods, making them more practical for scaling to larger datasets like ImageNet.

---

> > ### Comment · Reviewer_24Ct · 2025-04-02
> >
> > I have thoroughly read the authors' response. They addressed my concerns well, so I have raised my score to 4.

---

> > > ### Author Response · Authors · 2025-04-08
> > >
> > > Dear Reviewer 24Ct,
> > >
> > > Thank you for taking the time to carefully read our response and for your constructive feedback throughout the review process. We’re glad to hear that our clarifications addressed your concerns, and we truly appreciate your updated evaluation.

---

### Official Review · Reviewer_rCkP · 2025-03-14

**Overall Recommendation:** 4

**Summary:**

This paper proposes a mode-guided diffusion model that leverages a pre-trained diffusion model for dataset distillation, eliminating the need for fine-tuning with distillation losses. The proposed approach enhances dataset diversity through three key stages:

- Mode Discovery – Identifies distinct data modes to ensure comprehensive dataset coverage.
- Mode Guidance – Encourages intra-class diversity by guiding samples to different modes.
- Stop Guidance – Mitigates artifacts in synthetic samples that could negatively impact model performance.

The method is evaluated on multiple datasets, including ImageNette, ImageIDC, ImageNet100, and ImageNet-1K, demonstrating accuracy improvements of 4.4%, 2.9%, 1.6%, and 1.6%, respectively, over state-of-the-art techniques. The approach significantly reduces computational costs by eliminating the need for fine-tuning diffusion models with distillation losses while maintaining or improving dataset quality and representativeness.

## update after rebuttal
I think the author's experiments addressed my concern, so I decided to raise my score to 4.

**Claims And Evidence:**

see strength & weakness

**Essential References Not Discussed:**

no essential references need to be further discussed

**Experimental Designs Or Analyses:**

see strength & weakness

**Methods And Evaluation Criteria:**

see strength & weakness

**Other Comments Or Suggestions:**

Can the author provide experiments on the larger size of the model to show that the methods can be scaled up?

**Other Strengths And Weaknesses:**

## Strengths:
- Novelty and Contribution: The paper presents a novel approach to dataset distillation by leveraging pre-trained diffusion models without requiring fine-tuning or distillation losses. The proposed method introduces three stages (Mode Discovery, Mode Guidance, and Stop Guidance) to enhance sample diversity and quality, addressing key limitations in existing techniques.

- Significance and Impact: By eliminating the need for fine-tuning, the approach significantly reduces computational costs, making dataset distillation more accessible to researchers with limited resources. The method’s applicability to large-scale datasets (ImageNette, ImageIDC, ImageNet100, and ImageNet-1K) demonstrates its robustness and scalability.
- Empirical Validation: The reported improvements (e.g., 4.4%, 2.9%, 1.6%, and 1.6% accuracy gains over state-of-the-art methods) suggest the effectiveness of the approach. The evaluation of diverse datasets strengthens the claim that the method is generalized well.
- Clarity of Key Ideas: The paper provides a structured breakdown of the proposed method, distinguishing it from existing approaches in dataset distillation. The introduction of Mode Discovery, Mode Guidance, and Stop Guidance is well-motivated and logically presented.

## Weakness
- Small model: Experiments mainly on small size models and baseline for ImageNet are relatively low.

**Questions For Authors:**

see suggestion

**Relation To Broader Scientific Literature:**

no significant contributions to the broader scientific literature

**Theoretical Claims:**

see strength & weakness

---

> ### Author Rebuttal · Authors · 2025-04-01
>
> We appreciate the reviewers' recognition of our work as novel and acknowledging our work to address the key limitations of existing techniques without requiring any fine-tuning. We are also grateful that the reviewer highlighted the significance and impact of our approach, with mention of our work reducing the computational costs, thereby making the dataset distillation more accessible to researchers with limited resources. Additionally, we are pleased that the reviewer found our experimental validation thorough highlighting method's robustness and scalability, and found our method to be logically and clearly presented. Below, we provide answers to the reviewer's comments.
>
> ### Model Scaling Experiments
>
> In the main paper, we showed that our method achieved SOTA on ImageNet-1K (IPC-50) with ResNet18. As requested by the reviewer, we extended our experiments to larger models (ResNet-50 and ResNet-101) and showed consistent performance improvements across models of different scales.
>
> | Method         | ResNet-18        | ResNet-50         | ResNet-101         |
> | -------------- | ---------------- | ----------------- | ------------------ |
> | _Full Dataset_ | _69.8_           | _80.9_            | _81.9_             |
> | SR2$^2$L       | 46.8±0.2     | 55.6±0.3      | 60.8±0.5       |
> | G-VBSM         | 51.8±0.4    | 58.7± 0.3     | 61.0±0.4      |
> | RDED           | 56.5±0.1     | -                 | 61.2±0.4     |
> | EDC            | 58.0±0.2     | 64.3±0.2      | 64.9±0.2       |
> | D$^4$M         | 55.2±0.1     | 62.4±0.1      | 63.4±0.1       |
> | **Ours**       | **60.2±0.1** | **64.6±0.4** | **67.7±0.4** |
>
> **Table 1: Results on ImageNet-1K with IPC 50. '-' means not reported.**
>
> ### Clarification about ImageNet-1k Baselines
> Although the baseline accuracy may appear low in isolation, our method surpasses previous works on ImageNet. To provide proper context, we compare against the accuracy achieved when training the same network on the full dataset. For example, ResNet-18 reaches 69.8% accuracy with the full dataset, whereas our approach attains 86% of this performance while using only 3.9% of the data, demonstrating its efficiency and strong relative performance.

---

### Decision · Program_Chairs · 2025-05-01

**Decision:**

Accept (oral)

**Comment:**

The submission has been reviewed by 4 reviewers.

After the rebuttal, all reviewers acknowledged the merits of the manuscript and agreed that it meets the bar of ICML.

Essentially, the reviewers found the proposed method well-motivated, well-designed, and yields good results.

There is no basis to overturn the consensus. As such, the AC recommends the acceptance. Congrats!

Please, however, do account for the suggestions in the final version.